# Analysis of the Functionality of a Mobile Network of Sensors in a Construction Project Supervision System Based on Unmanned Aerial Vehicles

Michał Strach [1,*], Krzysztof Różanowski [2], Jerzy Pietrucha [3] and Jarosław Lewandowski [4]

1  Faculty of Geo-Data Science, Geodesy, and Environmental Engineering, AGH University of Krakow, 30-059 Kraków, Poland
2  Centre for Advanced Materials and Technologies CEZAMAT, Warsaw University of Technology, 02-822 Warszawa, Poland; k.rozanowski@gmail.com
3  Faculty of Organization and Management, Lodz University of Technology, ul. Wólczańska 221, 93-005 Łódź, Poland; jerzy.pietrucha@p.lodz.pl
4  Pietrucha International Sp. z o. o., 98-235 Błaszki, Poland; j.lewandowski@pietrucha.pl
*  Correspondence: strach@agh.edu.pl

**Abstract:** This manuscript presents the results of a project related to the construction and testing of selected devices included in a space inspection and worker supervision system. The most important components of this system are a swarm of unmanned aerial vehicles, a docking station for the automatic charging of many drones, monitoring sensors, and user software that integrates all components responsible for mission planning (UAV raids) and measurement data processing. All components were built according to an original solution. The main part of this manuscript is a description of tests used to verify the functionality of a sensor network for monitoring infrastructural elements and moving objects, including people working on a construction project. As part of this research, procedures for testing sensor networks under laboratory and field conditions were developed. The tests performed demonstrated the ability of the MESH network to self-organize depending on the location of the elements in the network. The system that was built ensured the transmission of data from telemetric devices during UAV flights, regardless of the coverage of terrain by other networks, such as Wi-Fi and GSM networks. Data were sent to the end user via a LAN network based on the IP protocol. The maximum range between devices forming the network and the range limitations caused by various terrain obstacles were also determined.

**Keywords:** mobile network of sensors; monitoring of employees; monitoring of infrastructural facilities; unmanned aerial vehicles; site supervision system





## 1. Introduction

Many decisions are directly or indirectly related to a location in space. Plans regarding the routes of new roads [1], the way of developing cities, the definition of new protected areas, and new locations of factories [2] or shopping centers all refer to precisely defined locations. The situation is similar with the sale or purchase of real estate [3,4], the monitoring of construction sites [5–7], and support for rescue operations [8–10]. Navigation systems are used while moving on land, on water, and in the air. Information about spatial locations is needed to estimate flood losses [11,12], conduct insurance risk analyses [13,14], analyze climate change [15,16], forecast the weather [17,18], and ensure public safety [19], as well as in defense [20,21]. Determining the locations and geometric relations of objects is possible through the use of various sensors, measurement systems, and advanced analytical software.

Work on the design, construction, and use of sensors and measurement systems is not limited only to research in laboratories. Sensors and related measurement systems occupy a large area in modern technology. Currently, most more or less complicated everyday

devices are equipped with control systems, an indispensable component of which are measurement systems with sensors. Progress in the development of mechanics, electronics, and computer science, as well as the increasing requirements of device users, has increased the importance of computer measurement systems. The use of computers enables the automation of measurement activities and the unattended operation of measurement systems.

Measurement systems that have been built and developed are used in many areas of social and economic life. One of the most dynamically developing economic sectors is the unmanned aerial vehicle (UAV) market. Unmanned aerial vehicles, which are also called drones, are becoming not only a modern but also an increasingly common tool in various areas of social and economic life. These devices can be divided into different types and categories. The most popular UAVs are multi-rotor devices [22]. This group includes devices with three, four, six, and even eight rotors. Multi-rotor devices are usually characterized by their small size, affordable price, ease of operation, and readiness for use. Their advantages also include the ability to take off and land vertically, as well as flight stability and hovering. The weakness of multi-rotor aircraft is their flight time, which is usually limited to 30 min. The second category of unmanned aerial vehicles includes helicopters, which are also known as single-rotor drones [23]. They are usually resistant to damage, can carry a heavier load, and can stay in the air longer. The next category is fixed-wing unmanned aerial vehicles, which are called aircraft [24]. A characteristic feature of their design is their one wing, and, to remain in the air, it is necessary for them to move. Fixed-wing drones require specialized training to operate and are often used in military operations.

The number of users of unmanned aerial vehicles is rapidly growing. The reason for this is the benefits and possibilities offered by this technology, especially in comparison with alternative data acquisition techniques, such as those using manned aircraft. The most important advantages of UAVs include their full readiness for work at any time, safety of use, universality of applications through the selection of appropriate sensors, and low cost in comparison with manned units. Their numerous advantages have caused drones to be more and more widely used at all stages of construction projects [25–28]. Sensors enable one to obtain appropriate materials for investors, designers, architects, and contractors.

The use of UAVs in the process of designing, building, and modernizing roads allows for the effective collection of data for such projects, as well as supervision of the construction process [29]. Drones are used in the construction industry for advertising and marketing purposes. Photographs of the project area may be used at the bidding stage. They are then used as a background for attractive architectural visualizations. Investors often order photographs and videos presenting completed projects and use these materials in marketing campaigns. Material obtained using unmanned aerial vehicles and made from hard-to-reach places or an unusual perspective attracts the attention of customers. For this reason, more and more materials shot with drones appear on the Internet or television.

Already at the project-planning stage, it is possible to prepare photographic documentation of an area for development. UAVs take photographs from the air and provide data for the construction of digital terrain models and orthophoto maps. The obtained documentation enables an analysis of the topography of the area intended for the project, as well as the verification of the scope of planned earthworks. Digital terrain models created from UAV excursions make it easier to determine the volume of excavations and embankments and allow for the dimensioning of foundations or the analysis of the depth of the foundation of underground cables. Digital data obtained from various sensors facilitate the accurate planning of the work's progress and the duration of a construction project. Moreover, working with digital data in a project's virtual environment allows for the elimination of any collisions that would generate additional work costs and delays at the construction site.

In recent years, there has been clear progress in the implementation of projects by using the BIM (building information modeling) methodology. Like any new technology,

the BIM methodology is implemented through pilot projects in public projects related to the construction of roads and railway lines. To apply this methodology, it is necessary to fill databases with clear and up-to-date information for the needs of all authorized participants in the project's process. Two elements are key in this process: the building object and its digital twin. Drones are responsible for providing information for the construction of a digital twin. A digital twin contains digital data about the terrain and its development, as well as 3D geometric parameters and the attributes of building objects at all stages of their life cycle [30,31].

In the construction process, unmanned aerial vehicles support the investment audit. Cameras mounted on unmanned platforms record images. The project's supervision inspector then has access to a view of hard-to-reach elements, e.g., bridges, flyovers, and roof coverings on buildings. This allows some irregularities during the construction of the infrastructure network and road construction layers to be located, the installation of lighting to be checked, the horizontal and vertical markings of routes to be determined, and small architectural objects to be visualized. The use of drones in multi-kilometer linear construction projects requires technical supervision over the works performed and attention to the appropriate quality of work; taking the applicable financial settlements into account is the optimal solution. Aircraft is used to continuously monitor the progress of work, people's activities, and the functioning of machines. Extended and detailed information about construction workers and the condition of infrastructure can be provided by using a network of sensors. These people-monitoring sensors provide information about the locations of individual employees in space, and they collect and analyze their vital functions (body temperature and pulse) and falls, e.g., after loss of consciousness [32]. Infrastructure-monitoring sensors are responsible for the permanent measurement of the condition and parameters of elements of the road and railway infrastructure and the accompanying devices [33–35]. Data from these sensors are transmitted through a self-organizing sensor network to a server [36,37]. Appropriate monitoring software analyzes and responds according to these measurement data.

Aircraft can reach most places on a road or railway line that is under construction in almost all weather conditions. Equipping a UAV with a thermal imaging camera also allows the observation of a construction site at night. Equipping unmanned aircraft with LiDAR allows data to be provided in the form of a point cloud. This technology supports the documentation of the shape and spatial location of natural and building objects. This is also the basis for analyzing the volume of earth masses and aggregates and inventorying geological forms. In addition, it can provide up-to-date information for the ongoing updating of models in BIM technology.

The above-mentioned capabilities of digital space inspection systems and the supervision of infrastructural projects significantly increase the level of safety on construction sites, enable the optimization of work, and save time and financial resources. The longest stage in the life cycle of road and railway infrastructure is its operation and maintenance. In this respect, the most important thing is to monitor infrastructure that is in use and supervise the safety of the transportation of passengers and goods. The use of sensors with high accuracy—especially laser scanners and digital cameras—provides accurate information about the analyzed elements. A dense point cloud enriched with high-resolution photographs can be the basis for assessing the technical conditions of road surfaces [38,39] and bridge structures [40]. Based on spatial data, it is possible to assess the technical condition of a railway traction network [41], determine the geometry of tracks and their connections [42], and analyze building gauges and geometric relations among objects. A dense cloud of points enriched with high-resolution photographs is the basis of the process of verifying and defining large lengths of railway lines.

Drones are used in road transport in many different ways, and research is still ongoing to expand the possibilities of their use. One of the applications is the recording of data with cameras mounted on UAVs to analyze road traffic and driver behavior. These data may be used, among other things, for supervision and monitoring, the recognition of road offenses,

assistance in managing road congestion, and the analysis of vehicle trajectories related to the assessment of accident risk [43–48].

An issue that has been particularly important in recent months is the monitoring of critical infrastructure during military, geopolitical, energy, and climate conflicts [49,50]. Military conflicts and terrorist attacks force us to prepare for possible threat scenarios. In such situations, institutions that manage critical infrastructure should be prepared to implement emergency plans and emergency response rules, as such decisions are urgent. In such situations, time is crucial when collecting up-to-date information about the locations and directions of the movement of people, the current conditions of the infrastructure, and possible sabotage activities.

It is impossible to characterize the possible applications of unmanned aerial vehicles and the related measurement sensors in a short study. For this reason, it was decided to present and analyze the functionality of a system that includes UAVs and other key components. This system was built for the needs of digital space inspection and project supervision using UAVs and mobile docking stations. This is a research and development project co-funded by the National Center for Research and Development in Poland. The practical goal of this project is to develop a system based on the following:

- A multi-rotor flying UAV platform—ultimately, this may be a UAV swarm of up to 16 devices [51];
- Automatic wireless charging stations based on multi-port docking stations;
- Measurement sensors located on moving objects (people and infrastructural elements) using an MWSN (mobile wireless sensor network);
- A U2U and U2I control and communication subsystem using an MANET (mobile ad hoc network);
- A geospatial description subsystem using GIS;
- EUS software, v. 1.09.

The developed system offers two categories of services. The first is a short-range monitoring service, which is based on flights performed in the line of sight with the required VLOS (visual line of sight) certification; multi-hectare areas can be monitored to identify threats and support the process of evacuation and rescue of victims of natural disasters based on these flights with the required BVLOS (beyond visual line of sight) certification. The second service is the digital documentation and supervision of the implementation of infrastructure projects. The main technological issue in this project is the achievement of the technical ability of a UAV platform to perform special tasks, e.g., carrying loads and supporting the activities of fire brigades and mountain rescue services in search-and-rescue operations. This work was preceded by design studies and laboratory tests. The individual components of the system were developed according to an original solution. The devices that make up the system can work independently, but their integration leads to a synergistic effect. Operating several unmanned aircraft at the same time makes their work much more efficient than that in operations performed with a single device. Mission support through automatic-docking charging stations multiplies the range of planned excursions, which is particularly important in the supervision of infrastructure projects. A network of sensors based on elements mounted on unmanned aerial vehicles increases the functionality of the construction project monitoring system. These features make the proposed system unique.

This manuscript describes the construction and principles of operation of a network of sensors used to monitor infrastructural elements and moving objects, including people working on a given construction project, in detail. The tested monitoring sensor system creates a MESH telemetry network. The network, through its self-organization, ensures the transmission of data from telemetric devices during the UAV's flight, regardless of the area covered by other networks, such as Wi-Fi and GSM networks. Data are transmitted to the end user via an IP LAN. As a result of this solution, the MESH network can be integrated with any network infrastructure.

This study presents procedures for testing sensor networks in laboratory and field conditions. Individual studies verified, among other things, the possibilities of network

self-organization, the maximum distance between devices, the impact of terrain obstacles on the operation of devices, and the remote monitoring of infrastructure and people. The results obtained here show the strengths and limitations in the functionality of this sensor network when monitoring construction projects.

## 2. Materials and Methods

The digital space inspection and project supervision system consisted of several key components. Each of them (the components) complemented the others, providing uniqueness and comprehensive capabilities for the entire system. Below is a brief description of the system components. This study focused primarily on the network of monitoring sensors, and tests were carried out to verify their functionality.

### 2.1. Unmanned Aerial Vehicles

The presented system consisted of several key components. The most important was an unmanned aircraft that was designed and built to fly in a drone swarm. It was assumed that it would be a multi-rotor device with the possibility of vertical take-off and landing in small and hard-to-reach areas. The solution that was designed was characterized by its great versatility and the ability to perform specific tasks. A big advantage was the ability to install various measurement sensors and position the UAV in a given position by hovering. The device had the following characteristics: a maximum remote control range (OcuSync Enterprise with CE transmission power) of up to 12 km with a real-time preview, a maximum mission range of up to 45 km (mission plan), a maximum flight time of up to 55 min, and a maximum altitude of 9000 m above sea level. Its maximum take-off weight (MTOW) was 24 kg, the payload could be up to 13 kg, and the flight speed could be up to 50 km/h with a wind resistance of 15 m/s. It was possible to plan work in the temperature range from –10 °C to 45 °C. The device's operation was completely isolated from the internet, and the system supported the MAVlink protocol. The UAV could make an emergency landing in the event of a propeller failure while maintaining horizontal and vertical control. The constructed platform was equipped with several permanent or replaceable mounted sensors. These included an ADS-B receiver (automatic surveillance and notification system for the aircraft's position), GEO-Aware, and a 4 MP 360° camera with 10× zoom (Figure 1).

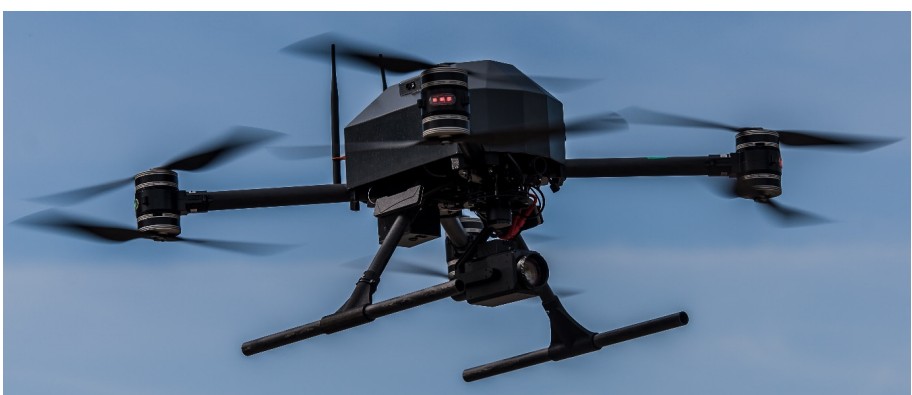

**Figure 1.** UAV model during the field tests.

The UAV platform was equipped with a gimbal on which sensors used for a specific purpose could be placed interchangeably. One of the sensors was a LiDAR sensor. To carry out photogrammetry tasks, it was equipped with a GPS RTK module, a precise IMU, and software enabling the processing of data from scanning measurement objects.

The next set of sensors were two professional digital photogrammetric cameras. They were equipped with video image stabilization, interchangeable lenses, and analog and digital zoom. The sensors also included a radiometric thermal imaging camera that allowed

the collection of accurate non-contact temperature measurements from a bird's eye view. The device recorded photos and videos from a height of up to 12 km.

The unmanned aerial vehicle was equipped with a system for detecting and avoiding obstacles and collisions (e.g., trees, power lines, etc.). The implemented technology enabled the stable performance of flight tasks such as hovering, movement along a given route, rotation, smooth take-off, and landing. The control technology ensured the high maneuverability of the UAV and the quick stabilization of the hovering platform when performing aerial thermal inspections in difficult weather conditions (e.g., strong winds, thick smoke, or fog). Ultimately, due to the set of specialized sensors and measurement sensors operating with various technologies (optoelectronic, acoustic, laser, and radar sensors) and the dedicated measurement and computation module, the system monitored and analyzed the environment, which will enable appropriate decisions to be made and alternative routes to avoid obstacles to be created.

Due to the use of LTE transmission and the planning of autonomous missions, the UAV became a long-range inspection platform with the ability to operate on difficult terrain. The applied flight control and mission-planning technology allowed the performance of completely autonomous missions according to a planned scenario. This can be applied to both a single UAV platform and several UAV platforms co-operating within one drone swarm.

### 2.2. Docking Station

A wireless automatic docking charging station provided a place for the UAV to land and be recharged directly in the task area. After docking, UAVs could be charged wirelessly. Energy for the charging system was supplied using standard power connections or through photovoltaic panels located on the docking stations. In addition to the basic functions of the station, it performed operations related to UAV support. The size of the constructed station allowed it to be transported to the location of a task by using an off-road vehicle or trailer. The size of the station allowed it to be loaded into the luggage compartment of a transport vehicle. The station's design was based on modular components, enabling efficient replacement in the event of damage and expansion of the station with additional functionalities. The station had a platform for UAV take-off and landing, as well as wireless and contact charging (Figure 2).

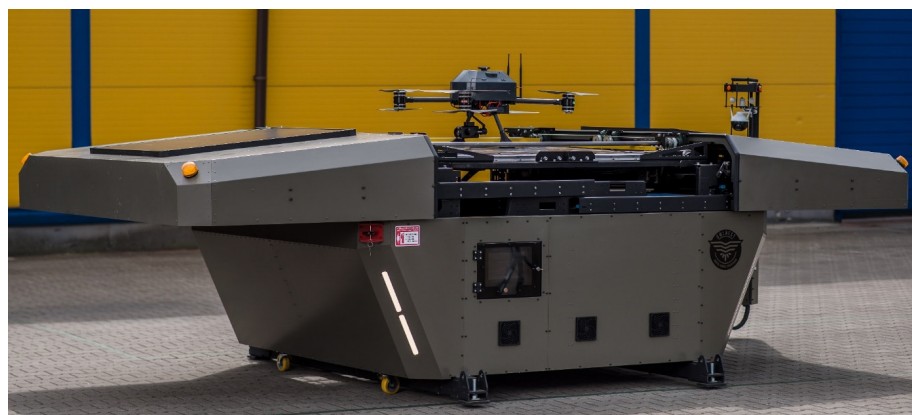

**Figure 2.** Charging station for automatic docking.

One of the advantages of wireless mobile charging stations for automatic docking is the multiplication of the range of a single UAV. Through this solution, the distance in planned missions is increased, which is particularly important in the process of monitoring multi-kilometer road projects.

### 2.3. Sensor Network

An important element of this system was the network of sensors for monitoring infrastructural elements and moving objects, including people working on construction sites. The tested monitoring sensor system created a MESH telemetry network. This network, due to its self-organization, ensured the transmission of data from telemetric devices during UAV flights, regardless of the area covered by other networks, such as Wi-Fi and GSM networks. Data were sent to the end user via IP LAN. Through this solution, the MESH network can be integrated with any network infrastructure. In practice, the router and the operator's computer may constitute one computer (logically, a server) in the ordering party's network (Figure 3).

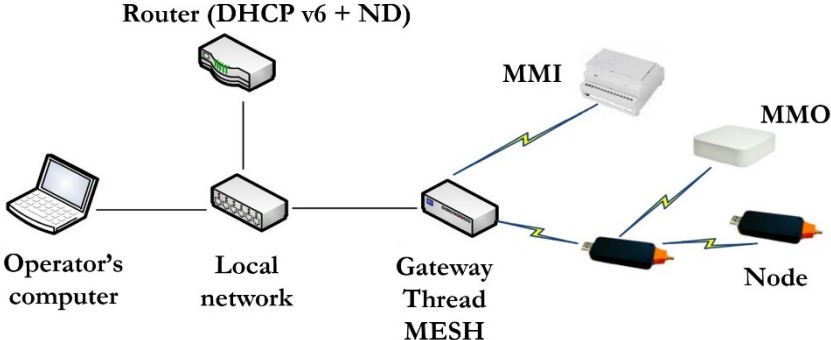

**Figure 3.** Proposed configuration of the MESH telemetry network.

The tested MESH network consisted of the following elements (Figure 4):

1.  Internal user network or computer: reading data from sensors, viewing employees' locations, and controlling actuators;
2.  Gateway: the gateway between the LAN and the telemetry system;
3.  Node–transceiver module: long-range module that creates a network bridge and enables the transmission of UAV telemetry data;
4.  MMO—person-monitoring module: location, measurement of environmental conditions (temperature and humidity), heart rate measurement, 9DOF sensor for fall detection, control of helmet wearing, and inductive charging;
5.  MMI—infrastructure-monitoring module: output control, reading digital and analog inputs, measuring physical values (temperature, vertical and horizontal deviations, distance changes, etc.), and three programmable digital interfaces.

The gateway was a maintenance-free device. It met the IP54 protection standard, which meant that it was resistant to dust and water splashes. To run the gateway, a power supply with an output voltage of 5 V (micro-USB) was required. The device came with a 2.5 A power supply with a micro-USB connector. The gateway had three USB interfaces and a LAN interface (Figure 5). USB connectors were optional and enabled connection between the gateway and, among other things, external memory. The fourth USB port was intended for service purposes. The gateway also had HDMI and audio interfaces, but, in the current configuration, they were inactive. Inside the device, there was a Raspberry Pi 3B+ microcomputer with a MESH module. Through this, the user could add their software to the gateway by installing it on a Raspbian system. There was a memory card on the side of the gateway. Optionally, the gateway could also be connected to a Wi-Fi network because it had a built-in WLAN and Bluetooth network card.

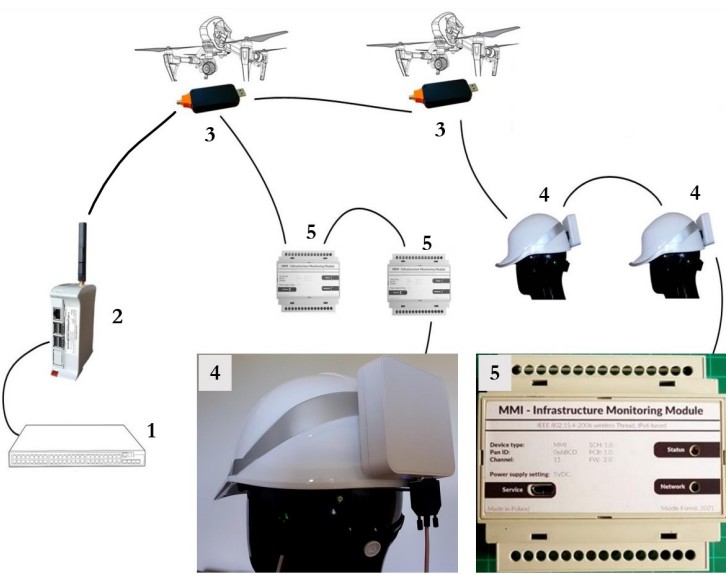

**Figure 4.** Concept of the MESH telemetry network's configuration.

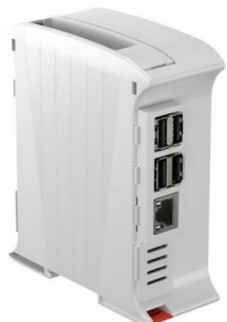

**Figure 5.** Gateway: the gateway between the LAN and the telemetry system.

The node was a maintenance-free device. It was connected to both unmanned aerial vehicles and docking stations. Due to its installation on a UAV, it was designed to have the smallest possible weight and dimensions. The node was equipped with an antenna screwed into the SMA socket (Figure 6). The device met the requirements of the IP54 protection standard. The node was visible in the system as a serial USB device. The node device needed to be connected to the USB port of a Raspberry Pi 3B+ computer (such a module was installed on the UAV and the docking station). After turning on the device, it automatically connected to the thread network.

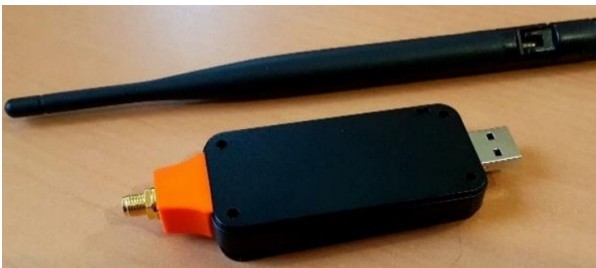

**Figure 6.** The thread network node with an antenna.

Another component of the system was the MMO person-monitoring module. This module was developed in a form that allowed the device to be mounted on a protective helmet (Figure 7a). There were no buttons on it. The goal was to prevent both tampering with the module and its being disabled by users. The buttons were located inside the

holes and could be pressed by using a thin object, such as a needle. The module had a built-in battery that could be charged from the micro-USB port or by using an inductive charger. The charger's receiving antenna was located in the center of the device on the front wall. The person-monitoring module was equipped with a fall sensor, a GPS receiver, and several sensors. These included the following sensors: those for temperature, humidity, atmospheric pressure, and heart rate based on a photo-optical sensor. The USB interface allowed the module to be charged and its software to be updated.

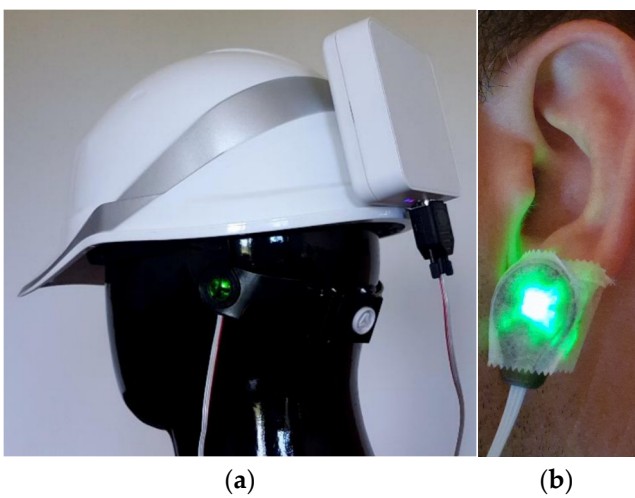

(**a**)  (**b**)

**Figure 7.** Example of mounting (**a**) the MMO module on a protective helmet and (**b**) a heart rate sensor on the earlobe.

An important addition to the person-monitoring module was the heart rate sensor. It was wired to the person-monitoring module (MMO). For research with the MMO, it was mounted on a protective helmet. The following factors influenced the installation of the MMO in this location:

- The minimization of the impact of obstacles on the reception of GPS satellite signals as a result of the height of the GPS receiver;
- The possibility of using an inductive charger without having to dismantle the module from the helmet (the receiving antenna was located on the front of the housing);
- The optimal location of the MESH network radio module (the antenna was located in a high position and was not covered);
- The comfort of wearing it (its being located on the helmet, unlike mounting on the wrist, pendant, or pocket, does not restrict movements, and has no contact with the skin; the MMO module is also less exposed to impacts and the false detection of falls).

The MMO was mounted on a protective helmet using industrial double-sided Velcro tape. An example of mounting the MMO module on a helmet is shown in Figure 7a.

The photo also shows a cable leading from the person-monitoring module to the heart rate sensor. In the proposed solution, the cable did not interfere with use. It also did not break down. During use, the front surface of the MMO module—behind which the antennae were located—was not covered by metal elements. The heart rate sensor was an external device. It allowed the measurement of the user's heart rate. The sensor used a clip equipped with a light source (green LED) and a detector—a phototransistor (Figure 7b)—for measurement. The shape of the designed housing was analogous to that of other heart rate sensors used in medicine. Heart rate sensors can only be mounted on certain areas of the body. These include the fingertips and toes (excluding the thumb and big toe), the earlobe, and other blood-supplied places where it is possible to observe the pulse of the flowing blood (temple and forehead). After testing with various mounting methods, it was found that gluing the sensor to the temple or earlobe would be the best. Therefore, a casing that could be mounted in both cases was made by using a dressing

plaster. This assembly seemed reliable, and it guaranteed correct measurement parameters. Medical sensors are installed in patients in a similar way. Dressing plasters protected the sensors from slipping off. In the case of the ear, the sensor could be equipped with a clip, but it could press against the earlobe and slip off during use. Pressure on the sensor would result in the inhibition of blood flow and a loss of the measurement capabilities.

The next element of the system was a stationary infrastructure-monitoring module (MMI). It was used to monitor infrastructural elements and enabled the connection of external sensors. The infrastructure-monitoring module was designed as a device that could be mounted on a DIN rail (Figure 4). A DIN rail is a standard metal-mounting rail used to mount modular electrical equipment. The constructed module needed to be mounted on a DIN rail in a control cabinet that guaranteed protection against weather conditions and direct heat sources. The module had a built-in battery that could be powered and charged using 5 V or 12 V direct current (DC). The MMI was equipped with interfaces for external sensors: UART, RS485, and CAN. The module also had two optically isolated digital outputs (delivered using optocouplers). The micro-USB interface allowed the internal battery to be charged and the software to be updated. The infrastructure-monitoring module enabled the wired connection of many external sensors. Their selection depends on the type of infrastructure and parameters that are subjected to permanent measurements in a given project.

### 2.4. Software

As part of the project, software that connected all project areas was developed. The most important functionalities included the following (Figure 8):

- Resource management (drones, users, and sensors);
- Flight planning, route preparation, and integration with QGroundControl;
- Management of the collected information, photographs, and sensor data;
- Monitoring of the sensors in the telemetry network.

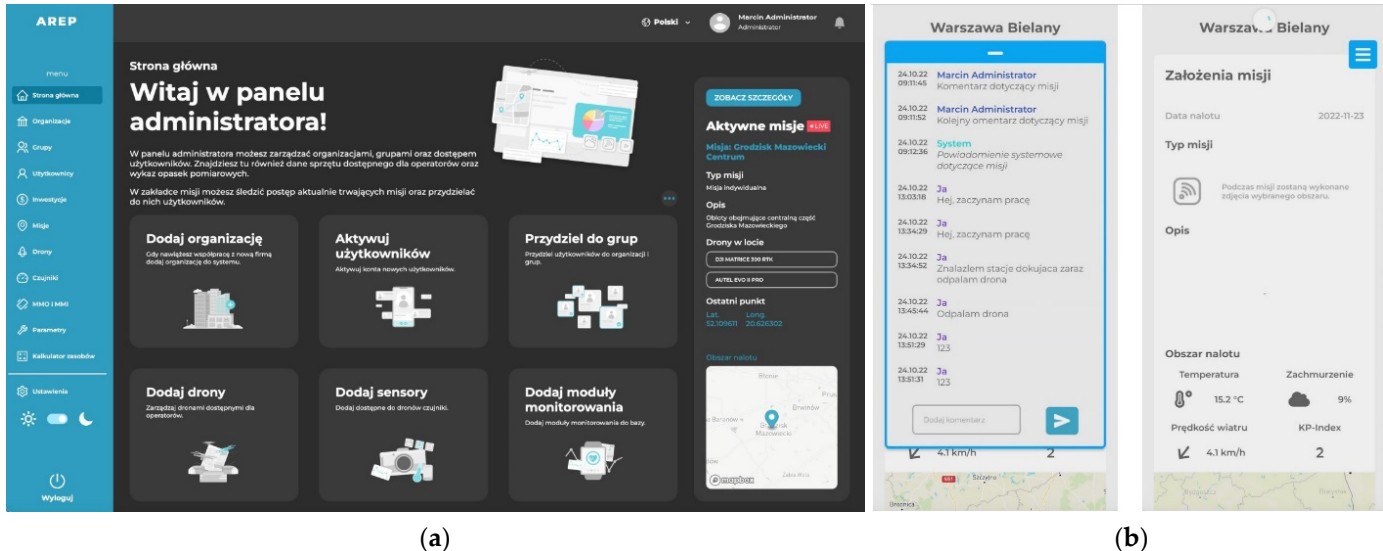

(**a**)

(**b**)

**Figure 8.** Application view: (**a**) web and (**b**) on mobile devices.

The design work also included the development of a geospatial GIS description subsystem, a mechanism for overlaying geodetic, design, and photogrammetric data, as well as digital terrain and infrastructure models, on the terrain's geodetic network. As part of the project, development work was carried out with the aim of precise parameterization of the process of monitoring water and civil engineering projects. Particular attention was paid to dangerous areas involving the use of unmanned aerial vehicles, as well as the

processing of geoinformation and cartographic data in a way that allowed for easy analysis of the project's process.

The system that was built also implemented security measures for data transmission and storage. Data transmission and the security of the information stored within the system were ensured in several areas. As part of the information exchange layer between the system components operating on data exchange in the L7 layer (application layer), an HTTPS connection with the TLS protocol version 1.3 was used according to the current IETF recommendations. In addition, the data architecture within the database assumed pseudo-anonymization at the level of the transmitted mission information. The security of the data stored in the database and data obtained in the process of carrying out missions was ensured with the use of an incremental backup mechanism, together with a mechanism implemented at the system level for the identification of data conflicts.

## 3. Tests of Monitoring Sensors Included in the MESH Telemetry Network

As part of this research, procedures for testing sensor networks in laboratory and field conditions were developed. The network's capabilities were analyzed in the following ways:

1. Self-organization depending on the locations of the sensors in the network;
2. System operation without access to the GSM network;
3. Receiving data from sensors without the need to build permanent networks;
4. Maximum distance between devices.

### 3.1. Laboratory Tests of the MESH Network

The first three options presented above were analyzed in laboratory tests. Appropriate devices and software were used in the tests. One of the devices was a sniffer whose task was to intercept data flowing in the network. It was a USB dongle that supported all short-range wireless standards. The nRF Thread Topology Monitor software was also used. It is a cross-platform tool that allows developers to visualize the topology of a thread MESH network in real-time. The research was carried out according to four scenarios related to the operation of the MESH telemetry network in laboratory conditions.

The first scenario assumed the arrangement of MMI sensors in such a way that each of them was within the range of the router (Figure 9a). As a result, each sensor established a direct two-way connection to the router. Additionally, all sensors were able to communicate through each other to the router as part of the MESH bridge. In the second scenario, the ability of the MMI sensors to build a MESH network was also tested. Using the network configuration that was previously prepared for the first scenario, one of the MMI sensors (MMI #2) was moved beyond the range of the router and the other sensors. The visualization in Figure 9b shows that MMI #2 was no longer an active device in the network (the module symbol is grayed out).

In the next scenario, the previously disconnected MMI #2 module was successively brought closer to the working MESH network. Over time, this module came within the range of one of the modules forming the network—in this case, MMI #1. This resulted in a one-sided connection between the modules and the reconnection of MMI #2 to the network (Figure 10). In the fourth and last scenario, the sensors were placed on a straight line. The distances between them were so large that they did not establish connections with each other. However, the sensors were connected in series in the MESH network (Figure 11). Individual modules sent data to the router as part of their work in their coverage circles. Laboratory tests were carried out, and the scenarios were used to confirm the ability of the constructed modules to build and self-organize MESH networks. It was also possible to operate the system without access to the GSM network, as well as to transmit and receive data between sensors.

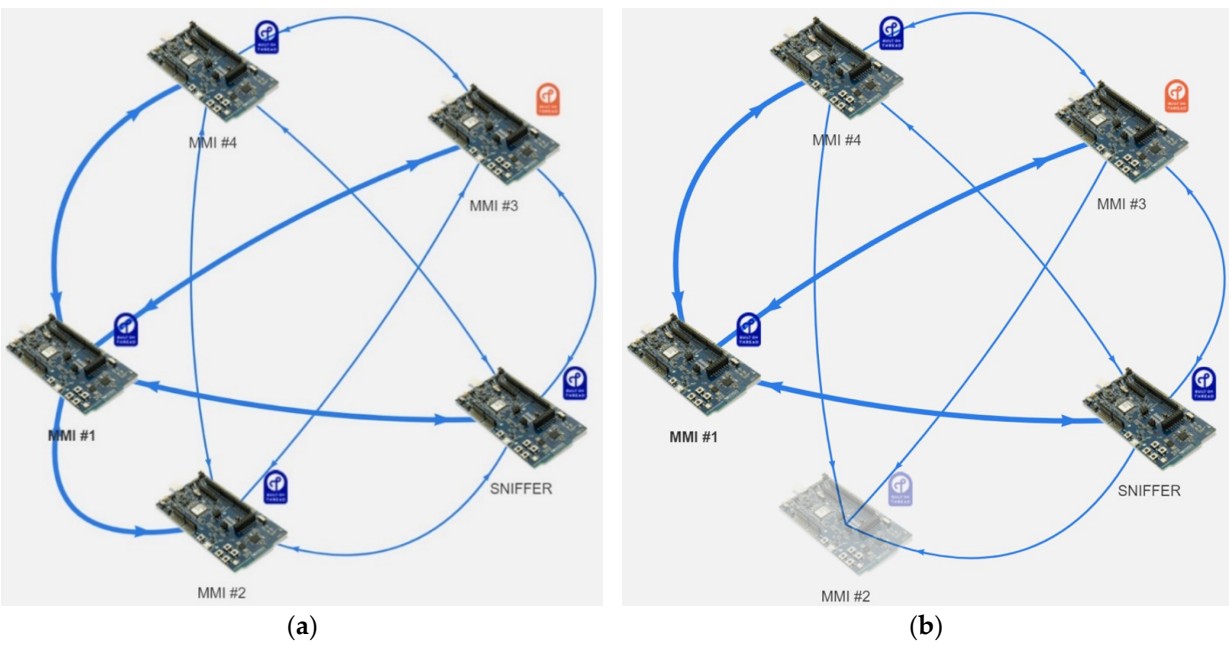

(**a**)                                                    (**b**)

**Figure 9.** Visualization of the MESH network scenarios: (**a**) all MMI sensors are within the range of the router; and (**b**) one of the sensors (MMI #2) is beyond the range of the router and other sensors.

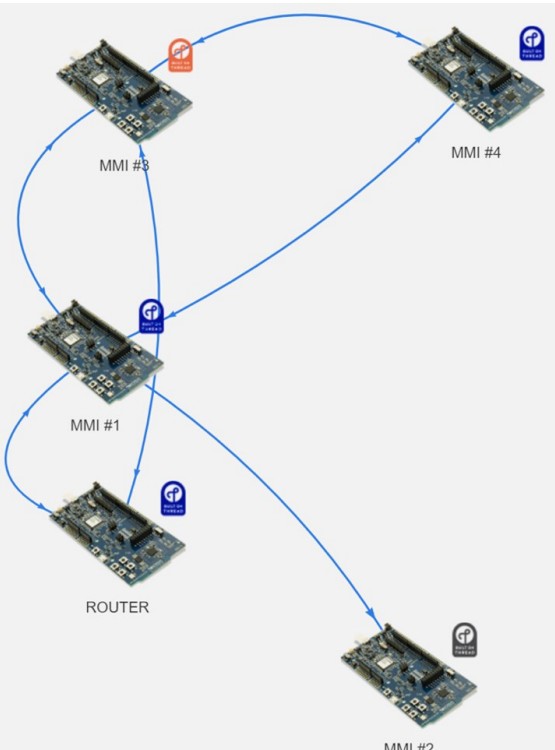

**Figure 10.** Visualization of the MESH network scenario in which MMI #2 was within the range of MMI #1 and established a successful connection to the network.

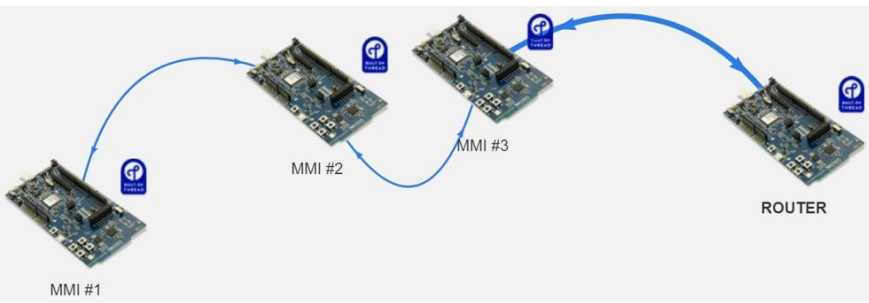

**Figure 11.** Visualization of a MESH network scenario in which serially connected sensors transmit data to the router as part of their operation within their coverage circles.

### 3.2. Field Tests of the MESH Network

Field tests were carried out in several places in Krakow (Lesser Poland Voivodeship, Poland). The selection of locations was guided by the varying intensity of the impact of terrain obstacles on the sensors. The first experimental site selected was in the northern part of Krakow, in the Pradnik Bialy district (Figure 12, area no. 1). The selected area was located in the flight corridor of passenger and military aircraft served by the Krakow–Balice airport. In that area, planes fly at an altitude of at least 388 m above ground level. There were ground navigation facilities for aircraft in the area. Single-family houses and apartment blocks were located around the research area. There was also a double high-voltage overhead power line running there.

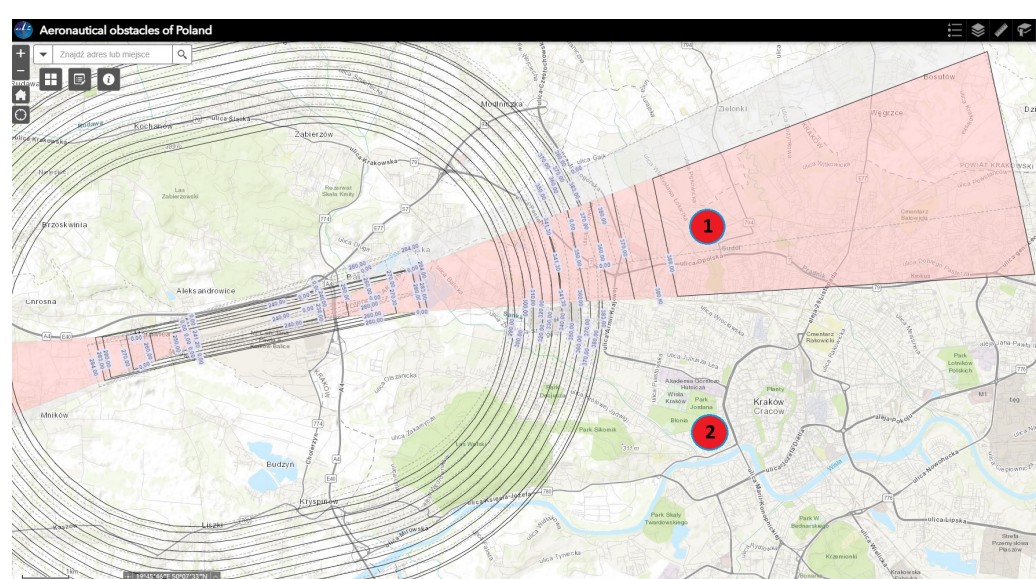

**Figure 12.** A map presenting the Kraków–Balice airport along with areas with a limited building height (OLS). The circles marked with numbers 1 and 2 are the locations of the experimental sites.

The second testing ground selected was in the Zwierzyniec district in Krakow's Blonia. Blonia is a city park in the form of a vast meadow with an area of 48 ha and a circumference of approximately 3700 m. This area was located west of the city center (Figure 12, area no. 2). In the selected area, there were no buildings or visible objects nearby that could affect signal propagation between the tested sensors.

Devices and sensors representing individual components of the MESH telemetry network were used in the tests. These were a computer, the gateway, the node, the MMO, and the MMI. The MMO and MMI sensors had an internal power source, but, if necessary, they could be connected to an external battery (power bank). The gateway was connected to the power bank each time. The node did not have its own power supply and was connected

to the other devices via USB. Depending on the type of testing, the node was connected to the gateway or a computer.

In the field tests, attempts were made to recreate the actual operating conditions of the MESH network. For this reason, the sensors were placed above the ground surface. Geodetic tripods were used in this research. Wooden tripods were selected for the highest precision in measurements. Wood is also a material that is neutral to radio waves, on which the MESH network worked.

### 3.2.1. Test of the Maximum Radio Range of the Node

The first test involved verifying the maximum radio range between node devices. Two nodes were used in this research. One of them was plugged directly into the USB port of the laptop. Ultimately, the carrier of the node would be unmanned aerial vehicles in place of the laptop. The second node was connected to the gateway, which was connected via Wi-Fi to the second laptop (Figure 13a). The laptop with the node installed was moved in the field. The tests were carried out several times in an area free from obstacles (Figure 12, zone 2). The signal between the devices was not interrupted. At the same time, the distance between the nodes was measured. For this purpose, a professional Leica GS16 GNSS receiver was used. The accuracy of determining the situational X and Y co-ordinates of this receiver was ±3 cm. Each time, the distances between the nodes exceeded 500 m. In Figure 13b, the distance measured in the tests was 514.04 m. If obstacles appeared between the node antennae, data transmission was interrupted. Transmission interference was influenced by buildings, groups of trees, and elevations in the terrain. After moving the node to a place without obstacles, transmission was established after about a dozen or so seconds. In the network functionality provided for in the project, the node devices will be installed on a UAV. Unmanned aerial vehicles perform missions at high altitudes, so the impacts of the above-mentioned obstacles will be minimized.

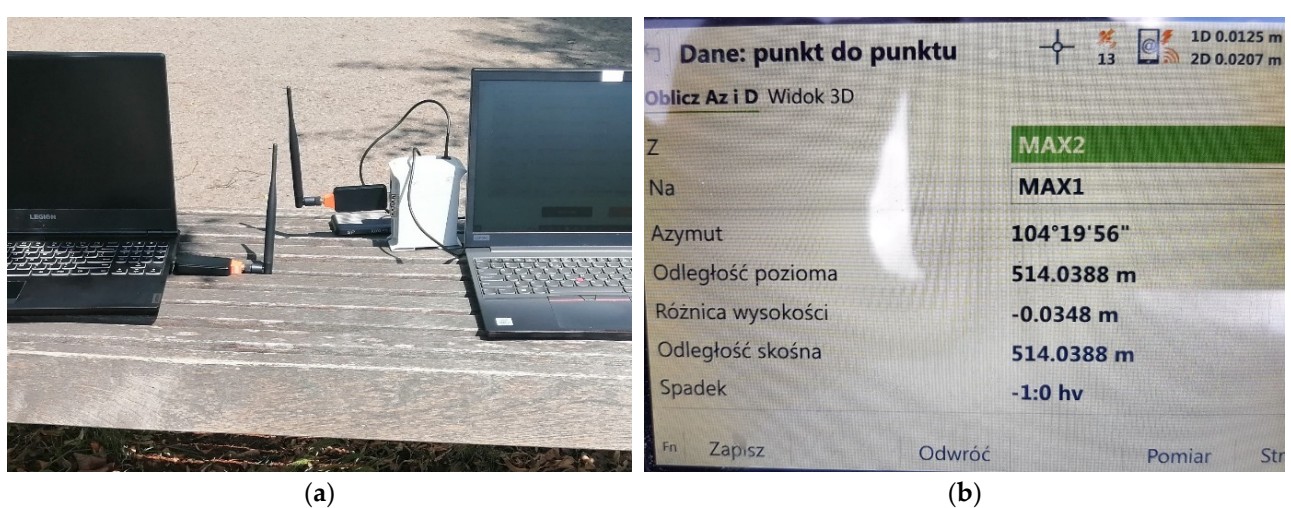

(**a**)                                                                (**b**)

**Figure 13.** Test of the maximum radio range of the nodes: (**a**) device configuration; and (**b**) display of the GNSS receiver controller with the result of the measured distance between the two nodes.

### 3.2.2. Tests of the Maximum Distance between Individual MESH Network Components

The next group of tests was to verify the maximum radio range between individual elements of the MESH network. The possibility of reorganizing the network, as well as the time needed to establish a connection between devices, was also checked. Each experiment was performed 2–3 times. The tests were carried out in zone no. 2 (Figure 12), which was free from terrain obstacles. Field tests were performed at different times under variable environmental conditions. One of the tests was performed at the turn of June and July. High air temperatures were recorded during this period. The tests began with full cloud cover, a storm passing nearby, light rainfall, and an air temperature of approximately 25 °C.

The tests were completed in sunny weather and a temperature of 29 °C. Further tests were performed in December in winter conditions at a temperature of 0 °C and under light freezing rain.

First, the range between the gateway and node was tested. In the studies, the distance between these devices was increased until radio communication was lost. Ultimately, the gateway maintained communication at a distance of 50–60 m, and, at such distances, the range of this device could be determined.

Then, the coverage of the MMO was verified. For this purpose, a node was connected to the gateway via a USB port. Then, the MMO module was moved away from the node, and the distance between them was measured. Ultimately, the MMO's range was determined to be 47 m.

Subsequent tests focused on determining the scope of the MMI. The test was analogous to the previous one. The gateway and node were used. At the other end of the section, there was the MMI module. The MMI's range was 55 m. It was, therefore, larger than the range of the MMO module.

The next range test concerned the gateway and MMI module. In the proposed configuration, the range of the MMI module was shortened to 45 m. It was proven that excluding the node shortened the reach of the MMO and MMI modules.

The last test in this series concerned the ability of the MESH network to create a serial connection between individual modules. The gateway, three MMOs, and two MMIs were used. Each device was placed on wooden geodetic tripods. Knowing the ranges of the individual modules, attempts were made to place them at distances that did not exceed 45–50 m (Figure 14). The MMI_5 infrastructure-monitoring module was installed at the end of the tested section. It was moved away until communication with the neighboring MMO_3 module was broken. Ultimately, a range of 95.1 m was achieved. This was the greatest range achieved by the MMI module in conditions free from radio signal interference.

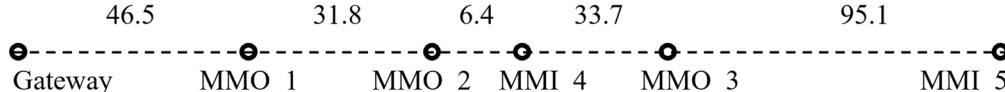

**Figure 14.** Sketch for testing the organization of a MESH network with a serial connection between modules (lengths given in meters).

All tests also analyzed the impacts of obstacles on data transmission interruptions. If there was an obstacle between the working devices, data transmission was interrupted. Even a person on the transmission line near one of the devices could be an obstacle. After removing the obstacle, transmission was established after about a dozen or so seconds.

The tests were performed several times under varying environmental conditions. Different ambient temperatures and freezing rain did not significantly affect data transmission or change the distance between devices.

### 3.2.3. Test of the Impacts of Obstacles on the Range and Operation of the MESH Network Components

The first test involved verifying the maximum radio range under high-voltage lines. Suitable testing conditions were found in zone 1, as shown in Figure 12. This was an area in which there was a double high-voltage overhead power line. The distance between the axes of both lines was 30 m. Wooden geodetic tripods were placed at both ends of the tested section (Figure 15). The gateway was set on one of them, and the node was connected to it. At the other end, there was an MMO module. The maximum range for the MMO module was 12.8 m. Then, a range test was performed for the MMI sensor. In this case, a range of 13 m was achieved. The results obtained proved that sources of interference, such as power grids, contributed to a significant reduction in the range of the MMO and MMI modules. The operating effectiveness of both types of modules decreased from 50 m to 13 m. In such an extremely unfavorable environment, the range of both sensors decreased by 74%.

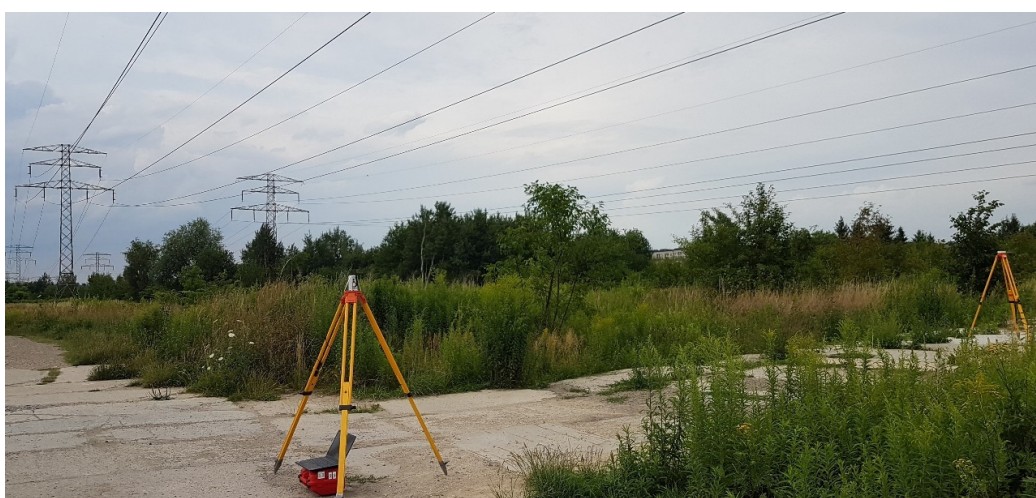

**Figure 15.** Range test of the MESH network modules under high-voltage overhead power lines.

The next test was a verification of the operation of the individual components of the MESH network when surrounded by terrain obstacles. The area around a steel warehouse hall was selected for this. The hall was 30 m long, 13 m wide, and 4 m high. The tests used the gateway, one infrastructure-monitoring module, and three person-monitoring modules (Figure 16).

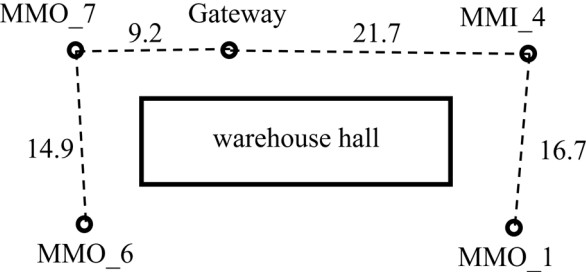

**Figure 16.** The UAV model during the field tests.

Launching individual modules resulted in the creation of a MESH network. Radio connections were organized by selecting the shortest path between individual devices. In Figure 16, a sketch of the connections between the devices is presented with a dashed line. All of the MMI and MMO modules transmitted data to the gateway, despite the lack of connection between MMO_1 and MMO_6. The distance between these modules was the largest and was over 30 m.

### 3.2.4. Test of the Heart Rate (Pulse) Sensor Available in the MMO Module

A test of the accuracy of the heart rate sensors was carried out by using two sets connected to MMO modules. The sensors were attached randomly to the left and right earlobes, respectively, with a band-aid (Figure 7b). The configuration of each of them allowed the heart rate to be recorded every 5 s. First, the MMO_1 heart rate sensor was connected to the left earlobe. The sensor recorded the pulse for 25 min of continuous operation. After some time, the MMO_2 heart rate sensor was connected to the right earlobe. It recorded the pulse for 15 min of continuous operation. For analysis, a reference pulse measurement was also performed on the person undergoing the tests. This measurement was performed twice—both before and after measuring the pulse at the earlobes. The reference measurement was made with an Omron M10-IT automatic upper-arm blood pressure monitor that provided the heart rate value. The option of averaging the results from three consecutive measurements was used. The reference result recorded with the

blood pressure monitor was stable and amounted to 52 beats/minute twice (both before and after the experiment).

The recorded results are presented in a line graph (Figure 17). Histograms were also created for the data recorded with both sensors (Figure 18). Additionally, calculations were performed to determine the basic statistical parameters of the datasets from both heart rate sensors (Table 1).

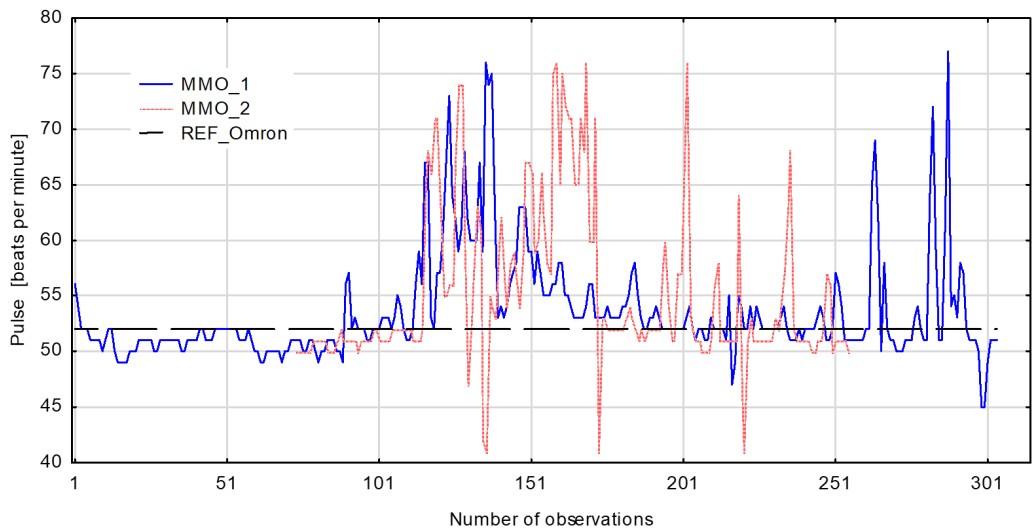

**Figure 17.** A line graph presenting the pulse recorded with both sensors—MMO_1 and MMO_2—as well as the reference device, an Omron M10-IT automatic upper-arm blood pressure monitor.

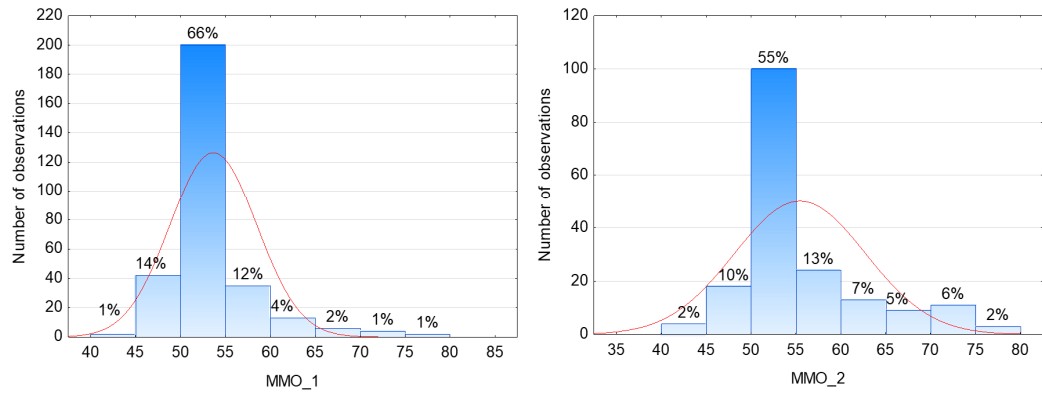

**Figure 18.** Histograms showing the pulse recorded with both sensors: MMO_1 and MMO_2.

**Table 1.** Basic statistical parameters for the datasets from the MMO_1 and MMO_2 heart rate sensors.

| Variable | Number of Observations | Mean | Median | Minimum | Maximum | Standard Deviation |
| --- | --- | --- | --- | --- | --- | --- |
| MMO_1 | 304 | 53.5 | 52.0 | 45.0 | 77.0 | 4.8 |
| MMO_2 | 182 | 55.3 | 52.0 | 41.0 | 76.0 | 7.2 |

Analysis of the graphs and statistical parameters allowed several conclusions to be drawn. The MMO_1 and MMO_2 sensors recorded the pulse for 25 and 15 min, respectively. The mean values of the recorded data were 53.5 and 55.3 beats per minute, respectively. The sample's standard deviation (*SD*) was also calculated according to the following equation:

$$SD = \sqrt{\frac{\sum_{i=1}^{n}(X - \overline{X})^2}{N - 1}}, \tag{1}$$

where *SD* is the sample's standard deviation, *X* represents each value, $\overline{X}$ represents the sample mean, and *N* represents number of values in the sample. The standard deviation values were 4.8 (MMO_1) and 7.2 (MMO_2), respectively. The average values were very similar to the results obtained from the measurement with the reference device, which was an upper-arm blood pressure monitor that gave the heart rate value. The blood pressure monitor recorded 52 beats per minute, and this was the median value for both tested sensors. The values of the standard deviations were influenced by single readings that were significantly different from the median. The minimum values reached 45 and 41 beats per minute, and the maximum values were 77 and 76 beats per minute. They constituted a small percentage of the recorded values, as illustrated in the line chart (Figure 17) and histograms, in which the percentage of the measured values was added (Figure 18). The reason for the outlying observations could be the movement of the subject's head, which could have resulted in slight rotations of the sensor on the earlobe while recording the heart rate. In the future, it is necessary to ensure a reliable way of mounting the pulse sensor and guaranteeing its stable position on the earlobe, regardless of the movements of the examined person's body.

3.2.5. Test of the Accuracy of the GPS Receivers Available in the MMO

Each person-monitoring module (MMO) was equipped with a GPS receiver. The test was aimed at verifying both the ability of the devices to create MESH networks and the accuracy of determining co-ordinates (location) with the GPS receivers installed in the person-monitoring modules. An area exposed to obstacles was selected for testing. There was a high-voltage network located several dozen meters away. Moreover, the research area was located in an aircraft flight zone. Wooden geodetic tripods were placed at the selected experimental site (Figure 19a). Six of them had one randomly selected (out of the 20 available) MMO. In the center of the test field, the gateway communication module and the long-range node module were placed on a tripod. The deployed modules worked properly and created a MESH network, and data from the devices were transferred and saved on the server. Observations were recorded at 10 s intervals for approximately one hour. The recorded data included, among other things, the following observations: the number of person-monitoring modules, date and time, atmospheric pressure, ambient temperature, longitude, and latitude.

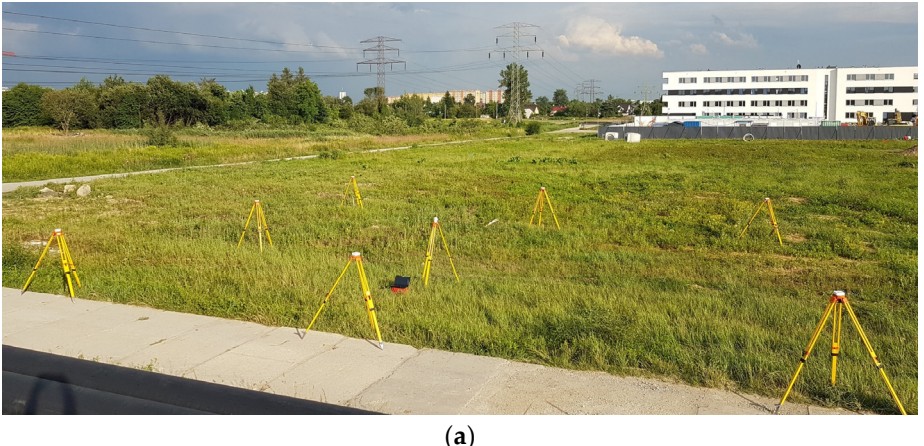
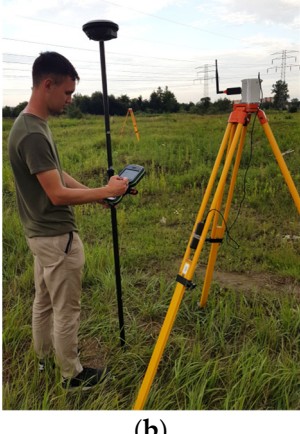

(**a**)          (**b**)

**Figure 19.** Testing ground for verifying the accuracy of the GPS receivers in MMO sensors (**a**); and (**b**) reference measurement of points with a professional GNSS receiver.

To analyze the accuracy of the GPS receivers in the MMOs, reference measurements were performed. For this purpose, a professional Leica GS16 GNSS receiver was used (Figure 19b). The accuracy of determining the situational X and Y co-ordinates with this receiver was ±3 cm. A GPS receiver was installed in the MMOs to determine the longitude

and latitude on the GRS-80 (WGS-84) ellipsoid: $(B, L)_{WGS-84}$. The values measured with each GPS receiver had to be transformed into the ETRF2000-PL/CS2000/21 co-ordinate system in which the Leica GS16 receiver operated: $(X, Y)_{2000/21}$. The conversions were performed according to the scheme shown in Figure 20.

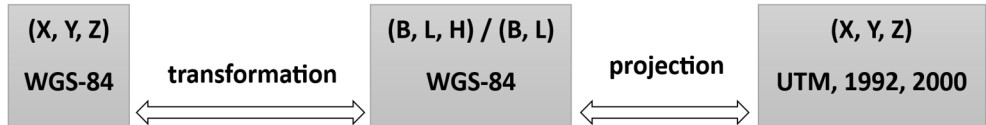

**Figure 20.** Scheme of the co-ordinate conversions and transformations.

A visualization of the location of the MMO sensor network and the recorded co-ordinates from their GPS receivers is presented in Figure 21a. The clusters of colored circles numbered GPS_1 to GPS_6 are subsequent records with the locations of MMO sensors. Duplicate co-ordinate values were removed for the purposes of visualization. An example sketch presenting the path of the apparent movement of the GPS receiver in MMO #3 is shown in Figure 21b. In the case of this receiver, 30 positions with non-repeating co-ordinates were selected from 280 observations. The white circles represent reference points determined with the precision Leica GS16 receiver. Thus, in Figure 21b, the reference point is described as GPS_REF 3.

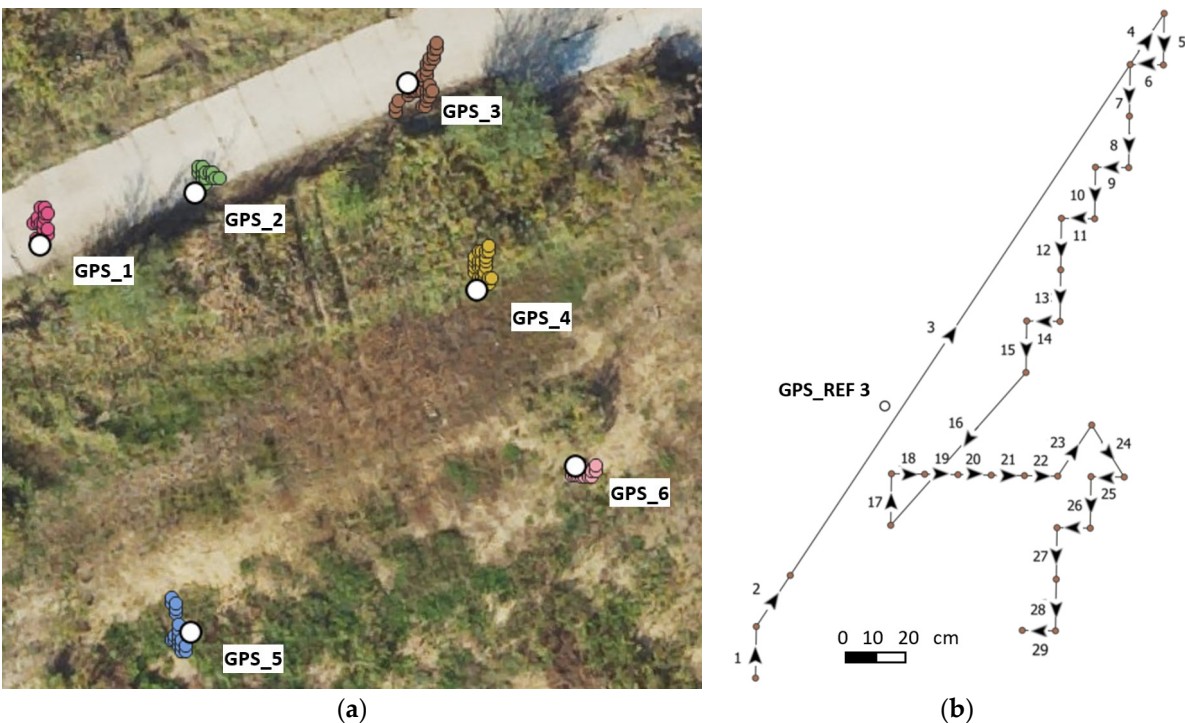

**Figure 21.** Visualization of the location of the MMO sensor network (colored circles) against the background of points measured with a precise GNSS receiver (white circles) (**a**) and (**b**) subsequent unique locations of the GPS receiver in MMO #3.

Based on the acquired co-ordinates, the distances between reference points determined with the Leica GS16 GNSS receiver and points determined with the six GPS receivers were calculated. The calculated values are illustrated in Figure 22. Additionally, calculations were performed to determine the basic statistical parameters for each of the six distance sets (Table 2).

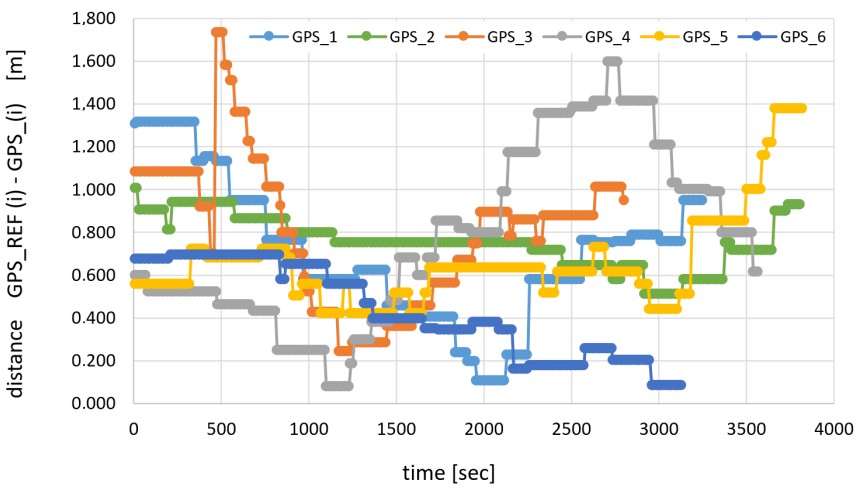

**Figure 22.** Line chart presenting the distances between the GPS_REF_(i) reference points determined with the Leica GS16 GNSS receiver and the points where the MMO sensors with GPS_(i) receivers were located.

**Table 2.** Basic statistical parameters for each set of distances [m].

| Basic Statistical Parameters | GPS_1 | GPS_2 | GPS_3 | GPS_4 | GPS_5 | GPS_6 |
|---|---|---|---|---|---|---|
| mean | 0.707 | 0.754 | 0.802 | 0.778 | 0.657 | 0.432 |
| median | 0.753 | 0.752 | 0.875 | 0.801 | 0.634 | 0.396 |
| standard deviation | 0.327 | 0.118 | 0.334 | 0.411 | 0.216 | 0.212 |
| maximum distance | 1.316 | 1.004 | 1.730 | 1.595 | 1.378 | 0.694 |
| number of observations | 325 | 381 | 280 | 357 | 382 | 313 |

Based on the analysis of the graphs and the results of the statistical analyses, several conclusions could be drawn. The individual MMO sensors created a MESH network and transmitted measurement data at 10 s intervals. The total number of recorded observations ranged from 280 (MMO_3) to 382 (MMO_5). The maximum distances between the reference point and the points determined in the measurement session with the six GPS receivers ranged from 0.69 m (GPS_6) to 1.73 m (GPS_3). The average and median values for the distances from individual GPS receivers were similar, and the differences between them for a given MMO sensor did not exceed a few centimeters. The maximum standard deviation for the compiled distances was 0.41 m. At this point, it is worth recalling the idea of using a GPS receiver in the MMO sensors. They are intended to provide, for example, the location of people participating in an investment process. The maximum distance that could be treated as accurate for the tested GPS receiver was 1.73 m. This is sufficient accuracy for locating any person within the operating range of the MESH network.

3.2.6. Test of the Temperature and Atmospheric Pressure Sensors Available in the MMO Module

The next datasets recorded by the MMO sensors were those of temperature and atmospheric pressure. Each of the six sensors operating in the MESH network measured and transmitted information at 10 s intervals for about an hour. During the tests, reference measurements of temperature and pressure were also made using a laboratory sensor. The results of the temperatures recorded with the MMO modules are presented in Figure 23. During the tests, the lowest temperature ranged from 24.7 °C (TEMP_3 and TEMP_6) to 25.7 °C (TEMP_5), with a temperature reading from the reference sensor of 24.8 °C. The highest temperature ranged from 27.9 °C (TEMP_3 and TEMP_6) to 29.0 °C (TEMP_5), with a reference result of 28.5 °C. The MMO_2 (TEMP_2) module had the most accurate temperature measurements. The differences in the indications of this module in relation to

the reference measurements did not exceed 0.4 °C throughout the recording of the results. The MMO_5 module measuring temperature (TEMP_5) achieved the largest differences from the reference measurements. The maximum difference was 1.8 °C. For the MMO_5 module, the average value for differences—which was similar to the median—was 0.9 °C.

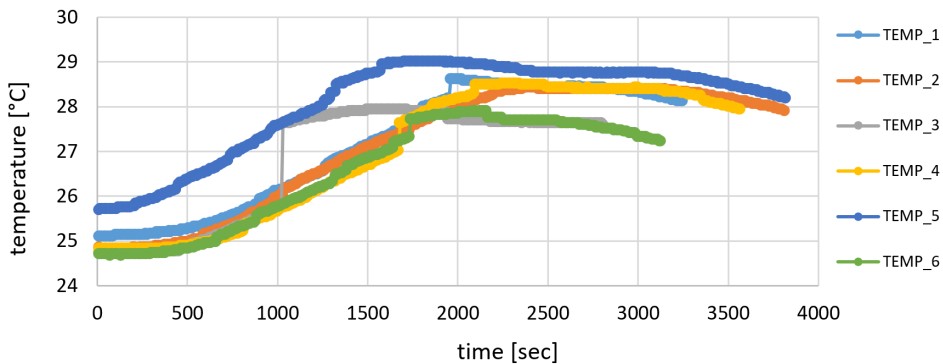

**Figure 23.** Line graph presenting the temperatures recorded with the MMO sensors.

All sensors recorded the temperature correctly. Only one temperature reading resulted in a maximum difference of 1.8 °C. Moreover, the differences in average temperatures between the tested sensors and the laboratory sensor did not exceed 0.9 °C. To monitor the ambient temperature in areas in which people operating a project are located, the determined accuracy of temperature measurements is sufficient.

Pressure measurements were also analyzed. Individual modules recorded the absolute pressure. This is the actual pressure that occurs on a given day in a given location. During the tests, the laboratory sensor showed a constant pressure of 988 hPa. Identical values were recorded by all six MMO modules. Only two of them—in 20% of cases (MMO_3) and 1% of cases (MMO_5)—recorded pressure jumps to a value of 989 hPa. It can, therefore, be concluded that the pressure measurements were recorded by all modules without any reservations.

## 4. Discussion

Tests of the digital space inspection and project supervision system using UAVs, mobile docking stations, and the MESH telemetry network proved that the technological integrity of the individual components was ensured. This study focused on verifying the functionality of the MESH telemetry network. The laboratory and field tests provided much valuable information about its functioning. Each network element operated correctly within its capabilities. One of the basic conditions for the system's operation was a short reorganization time for the MESH network's architecture. The tests confirmed the ability of the constructed modules to build and self-organize MESH networks in a few to several seconds. The system worked without access to GSM and Wi-Fi networks.

Another issue was the integration of the MESH network elements with UAVs—both mechanically and electrically. The model of the node module, which will ultimately be installed on an unmanned aircraft, was optimized in terms of weight and minimum energy consumption.

The tests of the maximum range of the nodes gave fully satisfactory results. In an area where there were no terrain obstacles, communication between the nodes exceeded 500 m.

Range tests between MESH network devices were also carried out. In the obstacle-free zone, the distance between the gateway and node was 50–60 m. The MMOs' range was 47 m, and the MMIs' range was 55 m. The exclusion of the node from the measurement set resulted in a reduction in the range of the MMO and MMI modules to 45 m. An interesting result was obtained in the tests verifying the ranges of the modules when a serial connection was created. The greatest range achieved by an MMI module was 95.1 m.

Limitations in the operation of the MESH network were observed in the vicinity of terrain obstacles and the vicinity of high-voltage lines. If an obstacle appeared between the node antennae or an MMI and MMO, data transmission was interrupted. Obstacles can be fixed or movable. If there was a person, vehicle, or other moving object in the line of data transmission, communication between the devices was interrupted. After removing the obstacle, transmission was established after about a dozen or so seconds. In the case of permanent obstacles, such as hills, buildings, and trees, transmission was not possible. The proximity of high-voltage overhead power lines also had a significant impact that limited the range and impeded the operation of the MESH network components. In such an unfavorable environment, the range of the MMO and MMI modules was shortened from 50 m to only 13 m.

It is worth mentioning that all kinds of obstacles had an impact on the range and disruptions in data transmission between elements of the MESH telemetry network. These included buildings, trees, and natural obstacles in the form of elevations in the terrain. An interruption of data transmission was also observed when there was a moving obstacle, such as a motor vehicle, a person, or an animal, in the line between MESH network devices. Range limitations must also be taken into account in the vicinity of power lines. In extremely unfavorable conditions, such as under high-voltage lines, the range decreased to only a dozen or so meters. Similar problems will need to be remembered when a node antenna is installed on a UAV. On the one hand, the high ceiling of unmanned aircraft minimizes the impacts of terrain obstacles on the range and data transmission. On the other hand, the high speed of a UAV with a node antenna on board will limit the measurement range. Fortunately, multi-rotor UAVs can be put into hover mode near operating MMOs and MMIs. Then, data transmission conditions will improve, and communication between network components will be established.

The MMO module enables the measurement of physiological and environmental parameters. The heart rate (pulse) was measured by using an optical sensor that was wired to the MMO. The average heart rate values were very similar to the results obtained from the measurement with the reference device. Outlying minimum and maximum heart rate values represented a small percentage of the recorded measurements. To obtain correct results, it must be ensured that the pulse sensor is firmly mounted on the earlobe or that the measurement is taken within the next few minutes while waiting for the measurements to stabilize.

The temperature and atmospheric pressure sensors worked fully and properly. The maximum difference between the reference measurements and those of the tested sensors was 1.8 °C. The measurements of atmospheric pressure with the sensors in the person-monitoring modules were identical to the measurements recorded with the laboratory sensor. In a small number of cases, two of the six sensors recorded atmospheric pressures that differed by only 1 hPa.

The GPS receiver included in the MMO equipment was mounted on a protective helmet. The maximum error of the position determined using the GPS receiver was 1.73 m. This is sufficiently accurate to locate a person within the MESH network's operating range.

The integration of infrastructure-monitoring sensors with buildings and selected technical devices remains to be solved. For this purpose, appropriate connectors will be selected soon; they will have to be mechanically durable and resistant to climatic conditions (IP65 protection level).

## 5. Conclusions

This summary emphasizes the contributions and innovative solutions of the digital space inspection and project supervision system. The following features are included.

- The beneficiary of a grant co-financed by the National Center for Research and Development in Poland has the full copyright for the construction and modification of each of the components of the system.

- The beneficiary has complete design documentation enabling the system to be built by any person or institution.
- The system was created based on original, innovative ideas for the construction and functioning of all its components.
- Each component was designed and built from scratch by a team of scientists and engineers in Poland.
- The system that was built is unique, and the use of its components leads to synergistic effects.
- The monitoring sensor system creates a self-organizing MESH telemetry network. This network ensures data transmission from telemetric devices during a UAV's flight, regardless of the areas covered by other networks, such as Wi-Fi and GSM.
- The constructed system is intended not only for research work but, above all, for commercial use in various missions. For this reason, the research carried out here and the results that were obtained are the basis for determining the functions of this system.
- The modular structure of the system allows the selection of hardware and software solutions to meet the user's needs.
- The system is characterized by a flexible method of operation by selecting sensors and adapting the operating mode to the current needs.
- The system is relatively cheap compared to the cost of renting manned aircraft with a team operating the equipment.

After being designed from scratch, built, and tested, the digital space inspection and project supervision system is ready to perform a variety of missions. The laboratory and field tests that were carried out showed both its strengths and limitations. The technical parameters of all components will have an impact on the functionality and, thus, the competitiveness of the constructed system. For this reason, it will be necessary to constantly improve the proposed solutions. The issues that will be addressed in the field of unmanned aerial vehicles include shortening the loading time and reducing the weights of sensors and cells that power drones while extending the mission duration, updating and selecting sensors and measurement systems to ensure more efficient operation while maintaining high precision in the measured values, and ongoing updating of the software controlling the operation of UAVs and sensors equipped with drones. A valuable solution is also the use of the potential of deep-learning applications in the device-to-device (D2D) communication of unmanned aerial vehicles (UAVs); tests and an implementation of this solution can be found in [52].

Automatic-charging docking stations have a modular structure, but they will be improved with modules that meet the needs of the largest possible group of recipients of measurement solutions. Ultimately, this also involves reducing the size and weight of docking stations while maintaining their existing functionality. These features will guarantee greater mobility of these devices.

The next component is the software responsible for the operation of individual components of the system. The functionality of this component will be influenced by updates to the software with new sensors and measurement systems, as well as the possibility of launching additional specialized missions. An equally important issue will be work on maintaining the security and resistance of the system to hacker attacks during missions.

The final component is the sensor network. The experiments carried out here showed the strengths of the constructed MESH network. However, one must be aware of the limitations associated with the use of individual network devices. This applies to both the distance between devices and the impacts of obstacles on range and data transmission. Nodes are installed not only on ground devices, but also on unmanned aerial vehicles. The spatial positions of drones [53] and the speed of their movement will have an impact on the range, data throughput, and data transmission rate.

The improvement of the digital space inspection and project supervision system described in this article using UAVs, mobile docking stations, and the MESH telemetry

network is partially dependent on external factors. The emergence of newer, more efficient sensors and measurement systems, as well as electronic components, will have an impact on solving some of the problems that were presented. The power limitations of the antennae responsible for transmitting signals are also important.

The motivation of the team of scientists and engineers working on creating the digital space inspection and project supervision system was its implementation in the widest possible range of applications. This team is responsible for research and development, within which it constantly creates, tests, and improves new technologies, products, and services. In its work, the team carries out experiments, analyzes data, tests prototypes, and implements new solutions on the market. For the commercial implementation of the system, co-operation with industries interested in its use is necessary. Due to its specificity, each industry may be interested in slightly different services and products. For this reason, it is essential to work with industry experts to individually determine the specifics of measurements and necessary products. The selection of the appropriate technology is influenced by several factors, including potential limitations/threats in the planning of air missions, the types and sources of data, their quality, the integration of data from various sensors, and the method of processing observations, as well as the form and method of sharing the results. Knowledge of the full information about the technological requirements for a given industry enables the individual selection of system components, planning of measurement missions, and subsequent development of final products. This approach translates into acquiring knowledge, skills, and competencies in the implementation of even the most complex work. This also translates into the uniqueness of the offer in terms of functionality and the possibility of the quick modification of the system components, as well as the variety of products obtained with their aid.

**Author Contributions:** Conceptualization, M.S. and K.R.; formal analysis, M.S., K.R., J.L. and J.P.; funding acquisition, K.R. and J.P.; investigation, M.S.; methodology, M.S., K.R. and J.L.; supervision, M.S.; writing—original draft, M.S.; writing—review and editing, M.S. The percentage of co-authors of a manuscript varies. The percentages are as follows: M.S.—70%; K.R.—10%; J.P.—10%; J.L.—10%. All authors have read and agreed to the published version of the manuscript.

**Funding:** This research was co-funded by the National Center for Research and Development in Poland (project no. MAZOWSZE/0055/19-00).

**Institutional Review Board Statement:** Not applicable.

**Informed Consent Statement:** Not applicable.

**Data Availability Statement:** The data presented in this study are available upon request from the corresponding author.

**Conflicts of Interest:** Jarosław Lewandowski was employed by the company Pietrucha International Sp. z o. o. The remaining authors declare that the research was conducted in the absence of any commercial or financial relationships that could be construed as a potential conflict of interest.

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
