# Peer review of "Analysis of the Functionality of a Mobile Network of Sensors in a Construction Project Supervision System Based on Unmanned Aerial Vehicles"

_sustainability, doi:10.3390/su16010340_

Round 1
Reviewer 1 Report
Comments and Suggestions for Authors
The paper presents the results of the project related to the construction and testing of selected devices included in the space inspection and investor supervision system. The most important components of this system are: a swarm of unmanned aerial vehicles, a docking station for automatic charging of many drones, monitoring sensors and user software integrating all components responsible for mission planning (UAV raids) and measurement data processing.
I think the article is well written and presents an acceptable idea realised in a real case.
I would just ask the authors to improve their summary by emphasising their motivation and contribution.
Furthermore, it would be preferable to draw up a table comparing their work with the most important articles in their literature.
The results presented are more or less well expressed but are not compared.
It is important to add equations, algorithms, etc. in order to enhance the value of the work.
the conclusion should be revised by adding the results.
Please quote the following articles
M. A. Ouamri, G. Barb, D. Singh, A. B. M. Adam, M. S. A. Muthanna and X. Li, "Nonlinear Energy-Harvesting for D2D Networks Underlaying UAV With SWIPT Using MADQN," in IEEE Communications Letters, vol. 27, no. 7, pp. 1804-1808, July 2023.
Ouamri, M.A., Singh, D., Muthanna, M.A. et al. Performance analysis of UAV multiple antenna-assisted small cell network with clustered users. Wireless Netw 29, 1859–1872 (2023).
Comments on the Quality of English Language
Moderate english correction required
Author Response
Thank you very much for undertaking the review of the manuscript. All comments, suggestions, and reservations constitute evidence of a detailed and professional analysis of the text of the article. Proofreading the text based on the Reviewers' suggestions will increase the scientific value of the manuscript. Some reviews note that moderate English editing is required. For this reason, the text of the manuscript was sent to MDPI Publishing House for English language editing. I am sending the corrected text for further analysis and review.
Below is my response to the Reviewer's comments.
Reviewer:
I would just ask the authors to improve their summary by emphasizing their motivation and contribution.
Answer:
A section has been added to the manuscript titled: “5. "Conclusions". There is a fragment responding to the Reviewer's suggestion:
The motivation of the team of scientists and engineers working on creating the digital space inspection and investment supervision system was its implementation in the widest possible range of applications. This team is responsible for research and development, within which it constantly creates, tests, and improves new technologies, products, and services. In its work, the team carries out experiments, analyzes data, tests prototypes, and implements new solutions on the market. For the commercial implementation of the system, cooperation with industries interested in its use is necessary. Due to its specificity, each industry may be interested in slightly different services and products. For this reason, it is essential to work with industry experts to determine the specifics of measurements and necessary products individually. The selection of work technology is influenced by several factors, including potential limitations/threats in the planning of air raids, type, and source of data, their quality, integration of data from various sensors, and the method of processing observations, as well as the form and method of sharing the results. Full information about technological requirements for a given industry enables individual selection of system components, planning of measurement missions, and subsequent development of final products. This approach translates into acquiring knowledge, skills, and competencies in the implementation of even the most complex works. This also translates into the uniqueness of the offer in terms of functionality and the possibility of quick modification of system components, as well as the variety of products obtained with its help.
Reviewer:
Furthermore, it would be preferable to draw up a table comparing their work with the most important articles in their literature.
Answer:
Studies of the scientific literature on the topics presented in the manuscript prove that it is difficult to find similar solutions to the described topic. The unmanned aircraft is characterized by parameters and equipment indicating high universality and efficiency. Additionally, it is possible to fly in a swarm of drones. The next component is the automatic landing docking station. There are few systems equipped with such efficient solutions. The last component is a self-organizing network of sensors for monitoring people and infrastructure. All components working together make the proprietary system unique.
Reviewer:
The results presented are more or less well expressed but are not compared.
Answer:
The built system is unique, so it is difficult to compare the obtained results with other measurement systems. The beneficiary of a grant co-financed by the National Center for Research and Development in Poland has full copyright to the construction and modification of each of the components creating the system. The beneficiary also has complete design documentation enabling the system to be built by a competent person or institution. The constructed system is intended not only for research work but above all for commercial use. For this reason, the research carried out and the results obtained are the basis for determining the functional functions of the developed and built system.
Reviewer:
It is important to add equations, algorithms, etc. in order to enhance the value of the work.
Answer:
Formula no. 1 and Figure 20 showing the scheme of coordinate conversions and transformations have been added to the text of the manuscript.
Reviewer:
the conclusion should be revised by adding the results.
Answer:
A summary of the results is presented in section: 4. Discussion. Additionally, the following part has been added: 5. Conclusions.
Reviewer:
Please quote the following articles
- A. Ouamri, G. Barb, D. Singh, A. B. M. Adam, M. S. A. Muthanna and X. Li, "Nonlinear Energy-Harvesting for D2D Networks Underlaying UAV With SWIPT Using MADQN," in IEEE Communications Letters, vol. 27, no. 7, pp. 1804-1808, July 2023.
Ouamri, M.A., Singh, D., Muthanna, M.A. et al. Performance analysis of UAV multiple antenna-assisted small cell network with clustered users. Wireless Netw 29, 1859–1872 (2023).
Answer:
In part: 5. Conclusions there are references to two articles. They have been added to References in positions 52 and 53. Thank you for pointing out these scientific works.
Reviewer 2 Report
Comments and Suggestions for Authors
The manuscript under review presents a comprehensive study on the construction and testing of a space inspection and investor supervision system, focusing on a swarm of unmanned aerial vehicles (UAVs), a docking station, monitoring sensors, and integrated user software. The ambitious project aims to enhance the efficiency and accuracy of monitoring infrastructure and moving objects, including construction site activities.
Strengths
1. The integration of UAVs with a MESH network for infrastructure monitoring is a forward-thinking concept.
2. The manuscript successfully describes the various components of the system, from hardware to software.
3. The field and laboratory testing procedures add practical validity to the research.
Drawbacks
1. The paper lacks a detailed analysis of the MESH network's reliability, particularly in adverse environmental conditions.
2. There is insufficient discussion on the security protocols for data transmission, a crucial aspect considering the sensitivity of monitoring data.
3. The manuscript does not address the system's scalability or provide a cost-benefit analysis, which is vital for potential real-world applications.
4. The user experience aspect of the software is not thoroughly explored, which is critical for end-user adoption.
Recommendations
1. A deeper exploration of the MESH network's performance under various conditions would strengthen the paper.
2. Incorporate a section on the security measures implemented for data transmission and storage.
3. Provide insights into the system's scalability and a detailed cost analysis to determine its economic feasibility.
4. Conduct user experience studies to evaluate the usability and intuitiveness of the software component.
The manuscript makes a significant contribution to the field of UAV-based monitoring systems. However, addressing the identified drawbacks and incorporating the suggested recommendations will enhance its practical applicability and academic robustness. The research presents a solid foundation for future studies and developments in this domain.
Author Response
Thank you very much for undertaking the review of the manuscript. All comments, suggestions, and reservations constitute evidence of a detailed and professional analysis of the text of the article. Proofreading the text based on the Reviewers' suggestions will increase the scientific value of the manuscript. Some reviews note that moderate English editing is required. For this reason, the text of the manuscript was sent to MDPI Publishing House for English language editing. I am sending the corrected text for further analysis and review.
Below is my response to the Reviewer's comments.
Reviewer:
Drawbacks
- The paper lacks a detailed analysis of the MESH network's reliability, particularly in adverse environmental conditions.
Recommendations
- A deeper exploration of the MESH network's performance under various conditions would strengthen the paper.
Answer:
The following fragments have been added to the manuscript:
3.2.2. Tests of the maximum distance between individual MESH network components
The next group of tests was to verify the maximum radio range between individual elements of the MESH network. The possibility of reorganizing the network, as well as the time needed to establish a connection between devices, was also checked. Each experiment was performed 2–3 times. The tests were carried out in zone no. 2 (Figure 12), which was free from terrain obstacles. Field tests were performed at different times under variable environmental conditions. One of the tests was performed at the turn of June and July. High air temperatures were recorded during this period. The tests began with full cloud cover, a storm passing nearby, light rainfall, and an air temperature of approximately 25 °C. The tests were completed in sunny weather and a temperature of 29 °C. Further tests were performed in December in winter conditions at a temperature of 0 °C and under light freezing rain.
A little further, abowe Figure 14:
The tests were performed several times under varying environmental conditions. Different ambient temperatures and freezing rain did not significantly affect data transmission or change the distance between devices.
Reviewer:
Drawbacks
- There is insufficient discussion on the security protocols for data transmission, a crucial aspect considering the sensitivity of monitoring data.
Recommendations
- There is insufficient discussion on the security protocols for data transmission, a crucial aspect considering the sensitivity of monitoring data.
Answer:
The following fragment has been added to the manuscript:
2.4. Software
…
The system that was built also implemented security measures for data transmission and storage. Data transmission and the security of the information stored within the system were ensured in several areas. As part of the information exchange layer between the system components operating on data exchange in the L7 layer (application layer), an HTTPS connection with the TLS protocol version 1.3 was used according to the current IETF recommendations. In addition, the data architecture within the database assumed pseudo-anonymization at the level of the transmitted mission information. The security of the data stored in the database and data obtained in the process of carrying out missions was ensured with the use of an incremental backup mechanism, together with a mechanism implemented at the system level for the identification of data conflicts.
Reviewer:
Drawbacks
- The manuscript does not address the system's scalability or provide a cost-benefit analysis, which is vital for potential real-world applications.
Recommendations
- Provide insights into the system's scalability and a detailed cost analysis to determine its economic feasibility.
Answer:
The disadvantages of the article include a lack of description of the system's scalability and a lack of cost-benefit analysis for real-world applications. There are several reasons why these issues were not presented. The first is the large size of the project and the resulting wide range of issues to be addressed. The authors decided that the leading issue would be the description of functional tests of the MESH telemetry network for monitoring infrastructure and people. The remaining system components are described very briefly. However, it was decided to present them so that the reader of the article could learn about the structure and functioning of the entire system. Another reason is the need to briefly present the selected research topic. Despite these limitations, the article is approximately 26 pages long. The last and equally important reason is related to the functioning of the beneficiary who received financial support. The project co-financing application submitted to the National Center for Research and Development in Poland includes a detailed project revenue model. This model assumes three modes of commercialization of research results: sales of the developed and built system, service, and making research results available in the form of licenses for results and construction documentation. This information constitutes the beneficiary's secret and therefore cannot be published.
Reviewer:
Drawbacks
- The user experience aspect of the software is not thoroughly explored, which is critical for end-user adoption.
Recommendations
- Conduct user experience studies to evaluate the usability and intuitiveness of the software component.
Answer:
The software for mission planning and the selection of measurement sensors installed on UAVs is intuitive and easy to use. Operating the MESH telemetry network is also not complicated. It is worth recalling what was written in the manuscript:
The MESH network, thanks to self-organization, ensures data transmission from telemetry devices during UAV flights, regardless of the area covered by other networks, such as Wi-Fi or GSM. Data is sent to the end user via IP LAN. Thanks to this solution, the MESH network can be integrated with any network infrastructure.
Additionally, both Gateway and Node are headless devices. The MMO people monitoring sensor and the MMI infrastructure monitoring sensor, after turning on the internal power supply, are capable of building a self-organizing MESH network. Measurement data from individual sensors are finally sent to the server (computer) and are ready for appropriate analyses. The developed MESH network was made available for testing to students of AGH University of Krakow. Conducting field tests and analyzing recorded observations was not a problem for them.
Additionally, a part was added to the manuscript called: “5. "Conclusions". There is the following fragment:
The improvement of the digital space inspection and project supervision system described in this article using UAVs, mobile docking stations, and the MESH telemetry network is partially dependent on external factors. The emergence of newer, more efficient sensors and measurement systems, as well as electronic components, will have an impact on solving some of the problems that were presented. The power limitations of the antennas responsible for transmitting signals are also important.
The motivation of the team of scientists and engineers working on creating the digital space inspection and project supervision system was its implementation in the widest possible range of applications. This team is responsible for research and development, within which it constantly creates, tests, and improves new technologies, products, and services. In its work, the team carries out experiments, analyzes data, tests prototypes, and implements new solutions on the market. For the commercial implementation of the system, cooperation with industries interested in its use is necessary. Due to its specificity, each industry may be interested in slightly different services and products. For this reason, it is essential to work with industry experts to individually determine the specifics of measurements and necessary products. The selection of the appropriate technology is influenced by several factors, including potential limitations/threats in the planning of air missions, the types and sources of data, their quality, the integration of data from various sensors, and the method of processing observations, as well as the form and method of sharing the results. Knowledge of the full information about the technological requirements for a given industry enables the individual selection of system components, planning of measurement missions, and subsequent development of final products. This approach translates into acquiring knowledge, skills, and competencies in the implementation of even the most complex work. This also translates into the uniqueness of the offer in terms of functionality and the possibility of the quick modification of the system components, as well as the variety of products obtained with their aid.
Reviewer 3 Report
Comments and Suggestions for Authors
A little bit too long introduction about UAVs.
The main problem is that you don't describe contributions or what is really novel. Please, bullet contributions.
There is no theoretical background. You only describe general informations about UAVs and specyfic informations about elements. But there are not to many details. This project is huge, but you assembly all parts together and tested it. Where is scientific soundness?
Your project is very interesting, but you must improve scientific/research area of it.
Point 4 - discussion should call "conclusions" or you must add it.
General - description of all is a little bit not science.
In fig. 4. pictures of real parts (MMI and MMO) are not swaped?
How about bad weather conditions? Tests was made in sunny day. What about rain?
Do you test communication with flying drone with high speed? Propably transmission distance will be shorter or transmission can be lost some time to time. You should say something about this issue (even only theoretical).
Comments on the Quality of English LanguageSome interpunction errors - commas, dots...
Little grammar correction is needed.
Author Response
Thank you very much for undertaking the review of the manuscript. All comments, suggestions, and reservations constitute evidence of a detailed and professional analysis of the text of the article. Proofreading the text based on the Reviewers' suggestions will increase the scientific value of the manuscript. Some reviews note that moderate English editing is required. For this reason, the text of the manuscript was sent to MDPI Publishing House for English language editing. I am sending the corrected text for further analysis and review.
Below is my response to the Reviewer's comments.
Reviewer:
A little bit too long introduction about UAVs.
Answer:
The large size of the project and the resulting wide range of issues to be addressed forced the authors to choose one leading research topic. The authors decided that the leading issue in the manuscript would be the description of functional tests of the MESH telemetry network for monitoring infrastructure and people. However, it should be emphasized that the developed and built MESH network is an integral part of the digital space inspection and investor supervision system. The most important components of this system are the drones and the automatic drone charging docking station. For this reason, it was decided to provide a broader description of the use of UAVs at all stages of construction investment implementation.
Reviewer:
The main problem is that you don't describe contributions or what is novel. Please, bullet contributions.
Point 4 - the discussion should end with "conclusions" or they should be added.
Answer:
Part 5. Conclusions have been added to the manuscript. It contains the following description:
This summary emphasizes the contributions and innovative solutions of the digital space inspection and project supervision system. the following features are included.
- The beneficiary of a grant co-financed by the National Center for Research and Development in Poland has a full copyright for the construction and modification of each of the components of the system.
- The beneficiary has complete design documentation enabling the system to be built by any person or institution.
- The system was created based on original, innovative ideas for the construction and functioning of all its components.
- Each component was designed and built from scratch by a team of scientists and engineers in Poland.
- The system that was built is unique, and the use of its components leads to synergistic effects.
- The monitoring sensor system creates a self-organizing MESH telemetry network. This network ensures data transmission from telemetric devices during a UAV’s flight, regardless of the areas covered by other networks, such as WiFi and GSM.
- The constructed system is intended not only for research work but, above all, for commercial use in various missions. For this reason, the research carried out here and the results that were obtained are the basis for determining the functions of this system.
- The modular structure of the system allows the selection of hardware and software solutions to meet the user's needs.
- The system is characterized by a flexible method of operation by selecting sensors and adapting the operating mode to the current needs.
- The system is relatively cheap compared to the cost of renting manned aircraft with a team operating the equipment.
After being designed from scratch, built, and tested, the digital space inspection and project supervision system is ready to perform a variety of missions. The laboratory and field tests that were carried out showed both its strengths and limitations. The technical parameters of all components will have an impact on the functionality and, thus, the competitiveness of the constructed system. For this reason, it will be necessary to constantly improve the proposed solutions. The issues that will be addressed in the field of unmanned aerial vehicles include shortening the loading time and reducing the weights of sensors and cells that power drones while extending the mission duration, updating and selecting sensors and measurement systems to ensure more efficient operation while maintaining high precision in the measured values, and ongoing updating of the software controlling the operation of UAVs and sensors equipped with drones. A valuable solution is also the use of the potential of deep learning applications in the device-to-device (D2D) communication of unmanned aerial vehicles (UAVs); tests and an implementation of this solution can be found in [52].
Automatic-charging docking stations have a modular structure, but they will be improved with modules that meet the needs of the largest possible group of recipients of measurement solutions. Ultimately, this also involves reducing the size and weight of docking stations while maintaining their existing functionality. These features will guarantee greater mobility of these devices.
The next component is the software responsible for the operation of individual components of the system. The functionality of this component will be influenced by updates to the software with new sensors and measurement systems, as well as the possibility of launching additional specialized missions. An equally important issue will be work on maintaining the security and resistance of the system to hacker attacks during missions.
The final component is the sensor network. The experiments carried out here showed the strengths of the constructed MESH network. However, one must be aware of the limitations associated with the use of individual network devices. This applies to both the distance between devices and the impacts of obstacles on range and data transmission. Nodes are installed not only on ground devices, but also on unmanned aerial vehicles. The spatial positions of drones [53] and the speed of their movement will have an impact on the range, data throughput, and data transmission rate.
The improvement of the digital space inspection and project supervision system described in this article using UAVs, mobile docking stations, and the MESH telemetry network is partially dependent on external factors. The emergence of newer, more efficient sensors and measurement systems, as well as electronic components, will have an impact on solving some of the problems that were presented. The power limitations of the antennas responsible for transmitting signals are also important.
The motivation of the team of scientists and engineers working on creating the digital space inspection and project supervision system was its implementation in the widest possible range of applications. This team is responsible for research and development, within which it constantly creates, tests, and improves new technologies, products, and services. In its work, the team carries out experiments, analyzes data, tests prototypes, and implements new solutions on the market. For the commercial implementation of the system, cooperation with industries interested in its use is necessary. Due to its specificity, each industry may be interested in slightly different services and products. For this reason, it is essential to work with industry experts to individually determine the specifics of measurements and necessary products. The selection of the appropriate technology is influenced by several factors, including potential limitations/threats in the planning of air missions, the types and sources of data, their quality, the integration of data from various sensors, and the method of processing observations, as well as the form and method of sharing the results. Knowledge of the full information about the technological requirements for a given industry enables the individual selection of system components, planning of measurement missions, and subsequent development of final products. This approach translates into acquiring knowledge, skills, and competencies in the implementation of even the most complex work. This also translates into the uniqueness of the offer in terms of functionality and the possibility of the quick modification of the system components, as well as the variety of products obtained with their aid.
Reviewer:
There is no theoretical background. You only describe general informations about UAVs and specyfic informations about elements. But there are not to many details. This project is huge, but you assembly all parts together and tested it. Where is scientific soundness?
Your project is very interesting, but you must improve scientific/research area of it.
General - description of all is a little bit not science.
Answer:
The large size of the project means that there is a very wide spectrum of interesting issues to be discussed. The authors decided that the leading issue would be the description of functional tests of the MESH telemetry network for monitoring infrastructure and people. The remaining system components are described very briefly. However, it was decided to present them so that the reader of the article could learn about the structure and functioning of the entire system.
The authors focused on a fairly detailed description of the elements and specifications of the devices creating the MESH telemetry network. It is worth adding that the beneficiary, through a multi-stage competition, received financial support for the construction of the system from the National Center for Research and Development in Poland. This is a scientific project reviewed and assessed both at the competition stage and during the final acceptance. The reviewers are independent scientific experts.
During the research, the beneficiary created construction documentation for all network components. He also holds the copyright to the construction and modification of individual system devices. The documentation constitutes the secret of the beneficiary and therefore cannot be published in the article.
I believe that the scientific value is demonstrated not only by the idea for the construction and operation of each component of the digital investment supervision system. The method of operation, laboratory and field tests carried out, as well as the analysis of the results of the devices forming the MESH network also have scientific value.
Reviewer:
In fig. 4. pictures of real parts (MMI and MMO) are not swaped?
Answer:
Figure 4 is correct. MMO people monitoring sensors and MMI infrastructure monitoring sensors may be mixed. This is possible thanks to the network's ability to self-organize. Moreover, each element of Figure 4 has been numbered. The numbering is correct.
Reviewer:
How about bad weather conditions? Tests were made in sunny day. What about rain?
Answer:
The following fragments have been added to the manuscript:
3.2.2. Tests of the maximum distance between individual MESH network components
The next group of tests was to verify the maximum radio range between individual elements of the MESH network. The possibility of reorganizing the network, as well as the time needed to establish a connection between devices, was also checked. Each experiment was performed 2–3 times. The tests were carried out in zone no. 2 (Figure 12), which was free from terrain obstacles. Field tests were performed at different times under variable environmental conditions. One of the tests was performed at the turn of June and July. High air temperatures were recorded during this period. The tests began with full cloud cover, a storm passing nearby, light rainfall, and an air temperature of approximately 25 °C. The tests were completed in sunny weather and a temperature of 29 °C. Further tests were performed in December in winter conditions at a temperature of 0 °C and under light freezing rain.
A little further, above Figure 14:
The tests were performed several times under varying environmental conditions. Different ambient temperatures and freezing rain did not significantly affect data transmission or change the distance between devices.
Reviewer:
Do you test communication with flying drone with high speed? Propably transmission distance will be shorter or transmission can be lost some time to time. You should say something about this issue (even only theoretical).
Answer:
In the manuscript in part 4. Discussion. The following description has been added:
It is worth mentioning that all kinds of obstacles had an impact on the range and disruptions in data transmission between elements of the MESH telemetry network. These included buildings, trees, and natural obstacles in the form of elevations in the terrain. An interruption of data transmission was also observed when there was a moving obstacle, such as a motor vehicle, a person, or an animal, in the line between MESH network devic-es. Range limitations must also be taken into account in the vicinity of power lines. In ex-tremely unfavorable conditions, such as under high-voltage lines, the range decreased to only a dozen or so meters. Similar problems will need to be remembered when a Node antenna is installed on a UAV. On the one hand, the high ceiling of unmanned aircraft minimizes the impacts of terrain obstacles on the range and data transmission. On the other hand, the high speed of a UAV with a Node antenna on board will limit the meas-urement range. Fortunately, multi-rotor UAVs can be put into hover mode near operating MMOs and MMIs. Then, data transmission conditions will improve, and communication between network components will be established.